# Batch Entanglement Detection in Parameterized Qubit States using Classical Bandit Algorithms

**K. Bharati**                                                          *ee20d700@smail.iitm.ac.in*
*Department of Electrical Engineering, IIT Madras*
*Chennai, India*

**Vikesh Siddhu**                                                          *vsiddhu@protonmail.com*
*IBM Quantum, IBM Research India*

**Krishna Jagannathan**                                                          *krishnaj@ee.iitm.ac.in*
*Department of Electrical Engineering, IIT Madras*
*Chennai, India*

**Reviewed on OpenReview:** *https://openreview.net/forum?id=0v27eMBVZO*

## Abstract

Entanglement is a key property of quantum states that acts as a resource for a wide range of tasks in quantum computing. Entanglement detection is a key conceptual and practical challenge. Without adaptive or joint measurements, entanglement detection is constrained by no-go theorems (Lu et al., 2016), necessitating full state tomography. Batch entanglement detection refers to the problem of identifying all entangled states from amongst a set of $K$ unknown states, which finds applications in quantum information processing. We devise a method for performing batch entanglement detection by measuring a single-parameter family of entanglement witnesses, as proposed by Zhu et al. (2010), followed by a thresholding bandit algorithm on the measurement data. The proposed method can perform batch entanglement detection conclusively when the unknown states are drawn from a practically well-motivated class of two-qubit states $\mathcal{F}$, which includes Depolarised Bell states, Bell diagonal states, etc. Our key novelty lies in drawing a connection between batch entanglement detection and a Thresholding Bandit problem in classical Multi-Armed Bandits (MAB). The connection to the MAB problem also enables us to derive theoretical guarantees on the measurement/sample complexity of the proposed technique. We demonstrate the performance of the proposed method through numerical simulations and an experimental implementation. More broadly, this paper highlights the potential for employing classical machine learning techniques for quantum entanglement detection.

## 1 Introduction

Quantum information theory has redefined quantum entanglement from a descriptive property of quantum states to a fundamental non-classical resource. As the basis for applications such as quantum communication, teleportation, and information processing (Bennett et al., 1993; Buhrman et al., 2001; Horodecki et al., 2009), entanglement detection and verification are central problems in quantum information science. Traditionally, this involves performing quantum measurements that yield probabilistic data, enabling techniques like full-state tomography (FST) for state reconstruction. However, this faces two challenges: theoretically, even after FST, determining entanglement remains computationally intractable, and this limitation is even more pronounced in scenarios involving many qubits; practically, real-world noise and imperfections limit the accuracy of state reconstruction. Modern multi-qubit compute systems may generate a bunch of entangled states across different sets of qubits through quantum gate operations; however, gate noise (e.g., phase flip errors, depolarization) can compromise their entanglement, requiring precise verification to ensure

their reliability before use in applications such as quantum computation and communication (Hong et al., 2010). In such contexts, FST can be employed for entanglement detection. However, it comes with a high computational burden, which may be unnecessary. We propose an alternative approach for simultaneously verifying or detecting entanglement among a given set of quantum states, dubbed *batch entanglement detection*. Instead of relying on FST, learning algorithms utilize statistical patterns to simultaneously analyze measurement data from a batch of quantum states and provide high probability guarantees on what they learn, i.e., whether or not the states are entangled.

Conventional techniques for learning quantum states include extensive research on FST (see Kueng et al. (2017); Wang et al. (2019); O'Donnell & Wright (2015a;b); Banaszek et al. (2013); Flammia et al. (2012) and references therein and also see Guta et al. (2020); Torlai et al. (2018); Quek et al. (2018); Koutný et al. (2022); Schmale et al. (2022); França et al. (2021) for machine learning-based approaches). Measurements required for FST scale exponentially with the number of qubits. While entangled measurements enable near-optimal copy complexity for FST (O'Donnell & Wright, 2015b; Haah et al., 2017), practical implementations rely on single-copy measurements, utilizing reconstruction methods such as linear inversion, maximum likelihood estimation, and maximum a posteriori estimation (Teo et al., 2011; Siddhu, 2019). These reconstructed states can be tested for entanglement using well-known criteria (some are outlined in Sec . 2.1). Alternatively, entanglement can be detected by measuring *entanglement witnesses* (Horodecki et al., 1996a; Terhal, 2000; Lewenstein et al., 2000a; Chruciski & Sarbicki, 2014), observables that detect *some* entangled states. No single witness can detect all entangled states, but in the worst case, combining information obtained from measuring different witnesses aids in state reconstruction via FST. This is explored in Zhu et al. (2010), where measurement operators from a family of six witnesses are used for bipartite qubit systems. The proposed approach for entanglement detection involves measuring a witness and formulating a *separability criterion* based on the frequencies of measurement outcomes. A negative value of the criterion indicates entanglement; otherwise, the process is repeated with another witness. If the state remains undetected by all witnesses, a tomographic reconstruction is performed (see Section 2.1 for further details).

Recently, the authors of Lumbreras et al. (2022) proposed using multi-armed bandit (MAB) frameworks for learning quantum states. The MAB algorithm repeatedly chooses from several options ("arms"), with the goal of finding the arm with the best outcome (the "best arm"). The algorithm balances between exploiting the known best options and exploring others to ensure no better option is missed (more details on MAB and policies can be found in Sec. 2.2). In Lumbreras et al. (2022), the inherent linearity in the quantum mechanical description of states is capitalized and a well-known classical learning algorithm that prescribes a sequential order of choosing measurements is employed. The MAB algorithms that are used provide guarantees on the quality of the estimate of the unknown quantum state. This MAB model in Lumbreras et al. (2022) does not directly apply to batch entanglement detection. Instead, it focuses on learning *one* entire quantum state, which may be unnecessary for entanglement detection.

Our first contribution builds on the witness-based separability criterion in Zhu et al. (2010) and uses suitable MAB policies for learning the same. We establish a formal connection between batch entanglement detection and the thresholding bandit problem (TBP) (Kano et al., 2018), enabling accurate and quick identification of $m$ entangled states from a batch of $K$ candidate states through adaptive measurement allocation. This formulation, which we refer to as the $(m, K)$-quantum MAB framework, differs structurally from the setting in Lumbreras et al. (2022) (see Remark 1) and focuses on learning entanglement-specific metrics without requiring full state reconstruction. Our second contribution uses classical MAB policies for adaptive measurement allocation and provides explicit measurement/copy complexity guarantees for batch entanglement detection–guarantees absent in FST and repeated witness testing Zhu et al. (2010). Using statistically guided confidence bounds, these policies are sample-efficient since they prioritise measurement effort only on uncertain states. Finally, we demonstrate the complete MAB-based pipeline for batch entanglement detection across multiple IBM Quantum backends, validate the framework under realistic noise and benchmark our results with non-adaptive tomographic approaches.

The rest of this paper is organized as follows: In Sec. 2, we provide a brief recap of some preliminary concepts in entanglement theory and multi-armed bandits. Readers interested in our connection between the two can move directly to Sec. 3, where we describe the $(m, K)$-quantum Multi-Armed Bandit framework for entanglement detection. We define a class of parameterized two-qubit states $\mathcal{F}$ and identify measurement

operators that conclusively detect entanglement in $\mathcal{F}$, detailed in Sections 4.1, 4.2, and 4.3. In Section 5, we demonstrate two TBP policies for entanglement detection. Section 6 analyzes the MAB policy performance on IBMQ backends and on an ibm-brisbane device for a family of states in $\mathcal{F}$ and details the quantum circuits used for simulation. Section 7 highlights measurement scheme limitations for entanglement detection in arbitrary states through numeric examples. In Section 8, we contextualize the numerical performance gains and discuss the practical advantages of the proposed MAB approach in comparison to existing state-of-the-art methods for entanglement detection like FST and fixed-witness testing (Zhu et al., 2010). Finally, Section 9 concludes the paper. Details on the non-adaptive tomography baseline and proofs of the results are presented in the paper and in Appendix A.

## 2 Preliminaries

Let $\mathcal{H}$ be a finite-dimensional Hilbert space with dimension $d$. A pure quantum state is represented by a unit norm vector $|\psi\rangle \in \mathcal{H}$. Let $\mathcal{L}(\mathcal{H})$ be the space of linear operators on $\mathcal{H}$, the Frobenius inner product for any $A, B \in \mathcal{L}(\mathcal{H})$, $\langle A, B \rangle := \text{Tr}(A^\dagger B)$ where $\dagger$ represents conjugate transpose. A Hermitian operator satisfies $H = H^\dagger$. A density operator $\rho \in \mathcal{L}(\mathcal{H})$ is Hermitian, positive semi-definite, $\rho \geq 0$, and has unit trace, $\text{Tr}(\rho) = 1$; it can represent both pure and mixed states. A positive operator value measure (POVM) is a collection of positive operators $\{E_i \geq 0\}$ that sum to the identity, $\sum_i E_i = I$. A POVM represents a measurement where $E_i$ corresponds to measurement outcome $i$, but sometimes we compress this and just say $E_i$ is a measurement outcome.

Let $\mathcal{H}_a$ and $\mathcal{H}_b$ be finite-dimensional Hilbert spaces with dimensions $d_a$ and $d_b$, respectively, and $\mathcal{H}_{ab} := \mathcal{H}_a \otimes \mathcal{H}_b$, where $\otimes$ represents tensor product, be a bipartite Hilbert space with dimension $d = d_a d_b$. A density operator $\rho_{ab} \in \mathcal{L}(\mathcal{H}_{ab})$ is called *separable* if it can be written as a convex combination of product states, that is,

$$\rho_{ab} = \sum_i p_i \left| \phi_a^i, \chi_b^i \right\rangle \left\langle \phi_a^i, \chi_b^i \right|, \tag{1}$$

where $p_i \geq 0$ such that $\sum_i p_i = 1$ and $\left| \phi_a^i, \chi_b^i \right\rangle := |\phi\rangle_a^i \otimes |\chi\rangle_b^i$ is a product of two pure states. We denote the convex set of all separable states by $S_{ab}$. Conversely, $\rho_{ab}$ is *entangled* if it can not be written in the form equation 1. We discuss some preliminaries on separability criteria for entanglement detection in Section 2.1 and background on stochastic multi-armed problems in Section 2.2.

### 2.1 Separability Criteria for Entanglement Detection

#### 2.1.1 Standard Analytical Separability Tests

Using full state tomography (FST), one can reconstruct any bipartite qubit state $\rho_{ab}$ and verify its entanglement through standard separability criteria (Horodecki et al., 2009). Notably, the Peres-Horodecki criterion (also called the PPT criterion) (Horodecki et al., 1996b; Peres, 1996) establishes that $\rho_{ab}$ is separable if and only if the eigenvalues of its partial transpose $\rho_{ab}^{\top_b}$ are non-negative. Here, $\top_b$ is the partial transpose with respect to $\mathcal{H}_b$. This condition remains necessary and sufficient for $(2 \times 3)$ systems but fails in higher dimensions due to the existence of bound-entangled PPT states. Other separability criteria include the range criterion (Horodecki, 1997), the matrix realignment criterion (Rudolph, 2000), the covariance matrix (CM) criterion (Gühne et al., 2007), and additional methods discussed in Gurvits (2003); Doherty et al. (2004).

#### 2.1.2 Entanglement Witness-based Separability Criterion

Entanglement can be detected by measuring entanglement witnesses, which can be defined as follows:

**Definition 1 (Entanglement Witness)** *An entanglement witness $W \in \mathcal{L}(\mathcal{H}_{ab})$ is a Hermitian operator satisfying,*

$$\langle \rho_{ent}, W \rangle = \text{Tr}(\rho_{ent} W) < 0, \quad \textit{for some entangled } \rho_{\text{ent}}, \tag{2}$$

$$\langle \rho, W \rangle = \text{Tr}(\rho W) \geq 0, \ \forall \rho \in S_{ab}. \tag{3}$$

Geometrically, a witness $W$ defines a hyperplane in the state space, delineating the set of *detectable entangled states* $D_W = \{\rho \text{ s.t. } \text{Tr}(\rho W) < 0\}$ from all separable states. For two arbitrary witnesses $W_1$ and $W_2$, $W_2$ is said to be *finer* than $W_1$ if $D_{W_1} \subseteq D_{W_2}$. A witness is said to be *optimal* when no strictly finer one exists, implying that it lies tangent to the boundary of the convex set $S_{ab}$ (Bengtsson & Zyczkowski, 2006). Further insights into this topology are detailed in Lewenstein et al. (2000b, Lemma 1).

**Single-parameter witness family:** We briefly review the witness-based separability criterion from Zhu et al. (2010). The authors propose a single-parameter witness family,

$$\rho_w(\alpha) = \cos^2 \alpha I - (|\psi\rangle \langle\psi|)^{\top b}, \tag{4}$$

where $|\psi\rangle = \cos\alpha |00\rangle + \sin\alpha |11\rangle$ such that $\alpha \in [0, \pi/4]$. We denote $\mathcal{E}$ to be the set of projectors onto the eigenstates of

$$\rho(\alpha)^{\top b} = (|\psi\rangle \langle\psi|)^{\top b} = \frac{1 + \cos 2\alpha}{2} |00\rangle \langle 00| + \frac{1 - \cos 2\alpha}{2} |11\rangle \langle 11| + \frac{\sin 2\alpha}{2} \big( |\Psi^+\rangle \langle\Psi^+| - |\Psi^-\rangle \langle\Psi^-| \big).$$

The set $\mathcal{E} = \{|00\rangle \langle 00|, |11\rangle \langle 11|, |\Psi^+\rangle \langle\Psi^+|, |\Psi^-\rangle \langle\Psi^-|\}$ forms a POVM and is referred to as a *Witness Basis Measurement* (WBM). For the remainder of the paper, we assume that the exact projective forms of the WBM are fixed and known.

**Quadratic WBM criterion:** The witness expectation value serves as a detection statistic, that is, $\text{Tr}(\rho W) < 0$ certifies entanglement, while non-negativity renders the test inconclusive. If the test is inconclusive for the base witness in equation 4, that is, $\text{Tr}(\rho_w(\alpha)\rho) \geq 0$, then subsequent witnesses are generated via local unitary transformations $U_1$ and $U_2$ as,

$$\rho_w(\alpha) \longrightarrow (U_1 \otimes U_2)^\dagger \rho_w(\alpha)(U_1 \otimes U_2). \tag{5}$$

with $(U_1, U_2) \in \big\{ (I, I), (I, X), (C^\dagger, C), (C^\dagger, XC), (C, C^\dagger), (C, XC^\dagger) \big\}$. Here, the operator $C$ cyclically permutes the Pauli operators $X$, $Y$ and $Z$, satisfying that $CX = YC$, $CY = ZC$, $CZ = XC$. Instead of performing a negativity test, the authors in Zhu et al. (2010) adopt a more stringent criterion:

$$\min_\alpha \text{Tr}\big\{ \rho_{\text{sep}} \big( \cos^2 \alpha I - \rho_w(\alpha) \big) \big\} \geq 0, \quad \forall \rho_{\text{sep}} \in S_{ab}. \tag{6}$$

The criterion is violated by the set of entangled states that *can* be detected by this witness family. The above optimisation leads to the following quadratic WBM criterion,

$$S = 4f_1 f_2 - (f_3 - f_4)^2 \geq 0, \quad \forall \rho_{\text{sep}} \in S_{ab}. \tag{7}$$

where $f_i := \text{Tr}\{E_i \rho\}$ are probabilities obtained from WBM $\mathcal{E}$. The value of $S$ in equation 7 depends on the underlying WBM. Thus, for a WBM $\mathcal{E}$ and state $\rho$, we denote equation 7 as $S_\mathcal{E}(\rho)$. We note that measuring the witness basis provides estimates for a distinct set of observables. For instance, the base witness in equation 4 yields estimates for three observables: $ZI + IZ$, $ZZ$, and $XX + YY$. The six-witness ensemble in total provides 15 independent expectation values, which provide sufficient information about the two-qubit state. Thus, if the state is undetected by the family of six witnesses, it can be reconstructed using these expectation values.

## 2.2 Fixed-Confidence Multi-Armed Bandit Policies

In this section, we briefly review some fixed-confidence policies for Best Arm Identification (BAI) and Good Arm Identification (GAI) in the stochastic Multi-Armed Bandit (MAB) setting, a canonical framework for sequential decision-making problems under uncertainty. A bandit instance $\boldsymbol{\mu}$ (problem instance) comprises $K$ arms, each described by a reward distribution $\nu_i$ supported on $\mathbb{R}$ with unknown mean $\mu_i$. In each round $t$, the learner chooses an arm $X_t$, receives an independent reward $Z_t \sim \nu_{X_t}$, and chooses the subsequent action based on a specified policy. We consider the class of $\delta$-correct policies, i.e., given a *fixed* error $\delta$ and

problem instance $\boldsymbol{\mu}$, such policies terminate in finite time and return the correct solution with probability at least $1 - \delta$. Mathematically, a $\delta$-correct policy $\pi = (\tau, \phi)$ satisfies:

$$\mathbb{P}^\pi_{\boldsymbol{\mu}}(\tau < \infty) = 1, \quad \mathbb{P}^\pi_{\boldsymbol{\mu}}(\phi = \phi_{\text{true}}) \geq 1 - \delta, \tag{8}$$

where $\tau$ is the stopping time, $\phi$ is the learner's final recommendation, and $\phi_{\text{true}}$ is the actual correct answer for the specific problem (BAI or GAI). Below, we briefly review well-known MAB policies that operate under *fixed-confidence* guarantees, aiming to make statistically reliable recommendations while minimising the number of samples.

### 2.2.1 Fixed-Confidence Best Arm Identification

In the BAI problem, the learner's objective is to identify the (best) arm $i^\star = \arg\max_{i \in [K]} \mu_i$ with the largest expected reward. Without loss of generality, arms are enumerated based on their expected reward $\mu_1 > \mu_2 \geq \cdots \geq \mu_K$ and the sub-optimality gap of arm $i$ is given by $\Delta_i = \mu_1 - \mu_i$. The performance of BAI policies is primarily characterised by the expected stopping time $\mathbb{E}_{\boldsymbol{\mu}}[\tau]$, which represents the expected number of samples required to recommend a best arm with confidence $1 - \delta$. The sample complexity improves progressively across algorithms: $\mathcal{O}(\Delta^{-2} \log(n\Delta^{-2}))$ for Successive Elimination (Even-Dar et al., 2002) and $\mathcal{O}(\Delta^{-2} \log \Delta^{-2})$ for LUCB (Kalyanakrishnan et al., 2012). Building upon this, Exponential-Gap Elimination (Karnin et al., 2013) and lil'UCB (Jamieson et al., 2014) utilize the law of the iterated logarithm (LIL) bounds to reach the near-optimal complexity of $\mathcal{O}(\Delta^{-2} \log \log \Delta^{-2})$, bridging the gap to the theoretical lower bounds (Farrell, 1964; Mannor & Tsitsiklis, 2004).

### 2.2.2 Fixed-Confidence Good Arm Identification

The GAI problem generalises BAI by introducing a threshold $\zeta$ and defining the set $\mathcal{G} = \{ i \in [K] : \mu_i \geq \zeta \}$ of "good" arms with unknown cardinality $|\mathcal{G}| = m$, leading to the $(m, K)$-GAI formulation. Without loss of generality, assume $\mu_1 > \mu_2 \geq \ldots \geq \mu_m \geq \zeta \geq \mu_{m+1} \ldots \geq \mu_K$ and the learner is unaware of this indexing. Fixed-confidence GAI policies adapt the notion of $(\lambda, \delta)$-*correctness* (Kano et al., 2018). A GAI policy is said to be $(\lambda, \delta)$-correct if, with probability at least $1 - \delta$, it correctly identifies at least $\lambda$ true good arms and does not misclassify any bad arm. Here, $\lambda$ specifies the number of correctly identified good arms. Unlike BAI, the sub-optimality gaps are denoted by $\Delta_i := |\mu_i - \zeta|$ and $\Delta_{i,j} = \mu_i - \mu_j$ and the sample complexity is expressed in terms of $\Delta = \min(\min_{i \in [K]} \Delta_i, \min_{j \in [K-1]} \Delta_{j,j+1}/2)$.

The goal, as in BAI, is to minimise the expected stopping time $\mathbb{E}_{\boldsymbol{\mu}}[\tau]$. However, a key difficulty in GAI is the exploration-exploitation dilemma of confidence, where the learner explores arms other than the empirical best arm to identify potentially 'good' arms with fewer pulls, while simultaneously exploiting the empirical best arm to increase confidence in its classification as a good arm. The Hybrid Dilemma-of-Confidence (HDoC) algorithm (Kano et al., 2018) combines UCB-based exploration (Auer et al., 2002) with LUCB-based elimination (Kalyanakrishnan et al., 2012), achieving sample complexity $\mathcal{O}\left(\Delta^{-2}\left(K \log \frac{1}{\delta} + K \log K + K \log \frac{1}{\Delta}\right)\right)$. The LIL-based refinement Tsai et al. (2024) lil'HDoC, employs tighter confidence widths to achieve $\mathcal{O}\left(\Delta^{-2}\left(K \log \frac{1}{\delta} + K \log K + K \log \log \frac{1}{\delta}\right)\right)$ samples, the best-known order for fixed-confidence GAI policies. The specific connections between BAI/GAI and entanglement detection are elaborated in Section 3 and 5.

## 3 The Quantum MAB Framework For Entanglement Detection

In this section, we introduce the quantum Multi-Armed Bandit (MAB) framework for batch entanglement detection. We formalise the structural similarity between the stochastic MAB model and its quantum analogue, where the learner interacts with a batch of quantum states by performing structured measurements.

### 3.1 Stochastic-Quantum MAB Mapping

In the stochastic MAB setting, pulling an arm $i$ corresponds to sampling from a probability distribution $p_i(\cdot)$ with known support and unknown mean $\mu_i$. Each pull yields a reward $j$ with probability (w.p.) $p_i(j)$ and

rewards across arm pulls are independent and identically distributed (i.i.d.). Analogously, in the quantum setting, each arm represents an unknown quantum state $\rho$. When $\rho$ is measured, the outcome distribution is determined by the fixed WBM. Specifically, if a WBM $\mathcal{E}$ is chosen, measuring $\mathcal{E}$ on $\rho$ will result in a outcome $j \in \{1, 2, 3, 4\}$ with probability $\text{Tr}(\rho E_j)$. Once the measurement is fixed, repeated measurements of $\rho$ yield i.i.d. outcomes. The key distinction lies in the source of the rewards: in the stochastic MAB model, rewards are sampled from classical distributions, whereas in the quantum MAB model, the rewards depend on i.i.d. outcomes obtained by measuring $\mathcal{E}$ on $\rho$.

## 3.2 Problem Setting and Objective

Given a batch of $K$ unknown quantum states $\{\rho_1, \ldots, \rho_K\}$, of which an unknown subset $m < K$ states are entangled, the learner's objective is to correctly identify all entangled states while minimizing the total number of measurements performed. Given a fixed WBM $\mathcal{E}$, the goal is to estimate the quadratic WBM criterion $S_{\mathcal{E}}(\rho_i)$ which indicates that $\rho_i$ is entangled if $S_{\mathcal{E}}(\rho_i) < 0$ and is inconclusive otherwise. The learner applies the MAB routine to this $(m, K)$ instance of quantum states under the chosen WBM $\mathcal{E}$ with the objective of accurately identifying $\mathcal{A}_{\text{ent}} = \{i \in [K] \text{ such that } S_{\mathcal{E}}(\rho_i) < 0\}$, using the fewest possible number of measurements. Since a single WBM may not detect all $m$ entangled states, and the value of $m$ itself is unknown, the MAB routine must be repeated for the six WBMs. Importantly, the measurement data collected under one WBM is *not* used to decide the next WBM; each WBM configuration should be treated as an independent instance. We summarise the stochastic-quantum MAB correspondence concisely in Table 1.

Table 1: Stochastic-Quantum MAB

| Attributes | Stochastic MAB | Quantum MAB |
|---|---|---|
| Arms | Probability distributions $(p_1, p_2, \ldots p_K)$ | Density operators $\{\rho_1, \rho_2, \ldots, \rho_K\}$ |
| Measurement | $-$ | WBM $\mathcal{E}$ |
| Measurement Data | $j$ w.p. $p_i(j), \forall i \in [K]$ | $j$ w.p. $\text{Tr}(E_j \; \rho_i), \; \forall j \in [4], \forall i \in [K]$ |
| Parameters to estimate | $\boldsymbol{\mu} = (\mu_1, \mu_2, \ldots \mu_K)$ | $\boldsymbol{S}_{\mathcal{E}} = (S_{\mathcal{E}}(\rho_1), S_{\mathcal{E}}(\rho_2), \ldots, S_{\mathcal{E}}(\rho_K))$ |
| Objective | Identify $\mathcal{G}^C = \{i \in [K] \text{ such that } \mu_i \leq \zeta\}$ | Identify $\mathcal{A}_{\text{ent}} = \{i \in [K] \text{ such that } S_{\mathcal{E}}(\rho_i) < 0\}$ |

We now formalise this correspondence by defining the $(m, K)$-quantum MAB setting.

**Definition 2** *The $(m, K)$-quantum Multi-Armed Bandit (MAB) setting for entanglement detection is fully characterized by the tuple $(\mathcal{A}, \mathcal{E})$. Here, $\mathcal{A}$ denotes a finite action set with $|\mathcal{A}| = K$, consisting of $(K - m)$ two-qubit separable states and $m$ two-qubit entangled states. The term $\mathcal{E}$ corresponds to a suitable Witness Basis Measurement (WBM).*

The objective of the $(m, K)$-quantum MAB problem for entanglement detection aligns with the classical $(m, K)$-Bad Arm identification where the goal is to find the set of "bad" arms $\mathcal{G}^C = \{i \in [K] \text{ such that } \mu_i \leq \zeta\}$ whose mean rewards fall below a threshold $\zeta$. Analogously, the $(m, K)$-quantum MAB problem seeks to identify the set of entangled states $\mathcal{A}_{\text{ent}}$ whose quadratic WBM scores $\boldsymbol{S}_{\mathcal{E}}$ violate the separability threshold. In essence, the $(m, K)$-Bad Arm identification setting and the $(m, K)$-quantum MAB problem for entanglement detection share a unified statistical structure, differing only in the interpretation of the reward model. To the best of our knowledge, this work is the first to establish a direct connection between stochastic MAB and quantum entanglement detection. This correspondence enables existing MAB algorithms to be directly applied in quantum settings, where the reward is encoded in the outcomes of WBM.

**Remark 1** *The d-dimensional discrete multi-armed quantum bandit model (Lumbreras et al., 2022) is different from our formulation. The authors consider arms to be a finite set of observables and the environment, an unknown quantum state $\rho$. The objective is to learn the unknown quantum state $\rho$ through an exploration-exploitation tradeoff. Given sequential oracle access to copies of $\rho$, each round involves selecting an observable to maximize its expectation value (reward). The information from previous rounds (history) aids in refining the action choice, thereby minimizing the regret, which is the difference between the obtained and maximal rewards. The authors also exploit the inherent linear structure in measurement outcomes and*

*map it to the linear bandit setting. Specifically, let $\{\sigma\}_{i=1}^{d^2}$ be a set of orthogonal Hermitian matrices. The unknown environment $\rho = \sum_{i=1}^{d^2} \mathrm{Tr}(\rho\sigma_i)\sigma_i = \sum_{i=1}^{d^2} \theta_i\sigma_i$ and arm $\mathcal{O}_t = \sum_{i=1}^{d^2} \mathrm{Tr}(\mathcal{O}_t\sigma_i)\sigma_i = \sum_{i=1}^{d^2} A_{t,i}\sigma_i$. Then, $\mathrm{Tr}(\rho\mathcal{O}_t) = \boldsymbol{\theta}^\top\mathbf{A}_t$ where $\boldsymbol{\theta} = (\theta_1, \theta_2, \ldots \theta_{d^2})$ and $\mathbf{A}_t = (A_{t,1}, A_{t,2}, \ldots A_{t,d^2})$. In round $t$, pulling arm $\mathcal{O}_t$ provides a reward $X_t = \boldsymbol{\theta}^\top\mathbf{A}_t + \eta_t$, where $\eta_t$ is 1-subgaussian.*

## 4 Parameterized Qubit States $\mathcal{F}$

To demonstrate the applicability of stochastic MAB policies for entanglement detection, we define a class $\mathcal{F}$ of parameterized two-qubit states defined as a union of three sub-families, $\mathcal{F} = \mathcal{F}_1 \cup \mathcal{F}_2 \cup \mathcal{F}_3$. These correspond to Depolarized Bell states ($\mathcal{F}_1$), Bell Diagonal states ($\mathcal{F}_2$), and amplitude-damped Bell Diagonal states ($\mathcal{F}_3$) whose explicit parametrisations are detailed in Sec 4.1, 4.2 and 4.3, respectively. We identify suitable WBMs from the witness family in equation 4 that can detect the same. We denote the first two witnesses in the witness family by $\mathcal{E}_1$ (base witness) and $\mathcal{E}_2$ (adapted using $(U_1, U_2) = (I, X)$), respectively. Here, $\mathcal{E}_1 \coloneqq \{|00\rangle\langle 00|, |11\rangle\langle 11|, |\Psi^+\rangle\langle\Psi^+|, |\Psi^-\rangle\langle\Psi^-|\}$ and $\mathcal{E}_2 \coloneqq \{|01\rangle\langle 01|, |10\rangle\langle 10|, |\Phi^+\rangle\langle\Phi^+|, |\Phi^-\rangle\langle\Phi^-|\}$.

### 4.1 Two-qubit Depolarized Bell States

For $w \in \mathbb{R}, -1/3 \leq w \leq 1$, a two-qubit **Depolarized Bell** state $\rho(w)$ is given by,

$$\rho(w) = w|\Upsilon\rangle\langle\Upsilon| + (1-w)\frac{I}{4}. \tag{9}$$

Here, $|\Upsilon\rangle$ represents any one of the four Bell states $|\Psi^\pm\rangle = (|01\rangle \pm |10\rangle)/\sqrt{2}$, $|\Phi^\pm\rangle = (|00\rangle \pm |11\rangle)/\sqrt{2}$. When $\Upsilon = |\Psi^-\rangle$, equation 9 is called a Werner state, and when $\Upsilon = |\Phi^+\rangle$, equation 9 is called an Isotropic state. The Peres-Horodecki criterion guarantees that $\rho(w)$ is separable when $-1/3 \leq w \leq 1/3$ and is entangled when $1/3 < w \leq 1$. Table 2 outlines the specific choices of WBM for the combination of the maximally mixed state with each of the four Bell states. When measured with these corresponding WBMs, the entangled depolarized Bell states are conclusively detected, determined by the value of $S = (w-1)^2/4 - w^2$ which is strictly positive for $-1 \leq w \leq 1/3$ and negative for $w > 1/3$.

Table 2: WBM for Depolarized Bell States

| Depolarized State | Pauli Basis | WBM |
|---|---|---|
| $w|\Phi^+\rangle\langle\Phi^+| + (1-w)I/4$ | $[I + \alpha(XX - YY + ZZ)]/4$ | $\mathcal{E}_2$ |
| $w|\Psi^+\rangle\langle\Psi^+| + (1-w)I/4$ | $[I + \alpha(XX + YY - ZZ)]/4$ | $\mathcal{E}_1$ |
| $w|\Psi^-\rangle\langle\Psi^-| + (1-w)I/4$ | $[I + \alpha(-XX - YY - ZZ)]/4$ | $\mathcal{E}_1$ |
| $w|\Phi^-\rangle\langle\Phi^-| + (1-w)I/4$ | $[I + \alpha(-XX + YY + ZZ)]/4$ | $\mathcal{E}_2$ |

### 4.2 Two-qubit Bell diagonal States

**Bell diagonal** states are a probabilistic mixture of the four Bell states. These states are more general than the ones in equation 9. Given parameters $p_1$, $p_2$, $p_3$ and $p_4$ such that $p_i \geq 0, \sum_i p_i = 1$, the Bell diagonal state is defined,

$$\rho_{\mathrm{BDS}} = p_1|\Phi^+\rangle\langle\Phi^+| + p_2|\Psi^+\rangle\langle\Psi^+| + p_3|\Psi^-\rangle\langle\Psi^-| + p_4|\Phi^-\rangle\langle\Phi^-|. \tag{10}$$

The eigenvalues of $\rho_{\mathrm{BDS}}^{\top_b}$ are calculated to be $1/2 - p_1$, $1/2 - p_2$, $1/2 - p_3$ and $1/2 - p_4$. Consequently, a Bell diagonal state is entangled if any one of these probabilities exceeds $1/2$, while the sum of the other three probabilities is less than $1/2$. Conversely, a Bell diagonal state is separable if all probabilities are less than or equal to $1/2$. Expressing equation 10 in the Pauli basis yields,

$$\rho_{\mathrm{BDS}} = \frac{1}{4}[I + aXX + bYY + cZZ],$$

where $a = p_1 + p_2 - p_3 - p_4$, $b = -p_1 + p_2 - p_3 + p_4$ and $c = p_1 - p_2 - p_3 + p_4$.

Table 3: WBM for Bell Diagonal States

| Probabilistic mixture | a | b | c | WBM |
|---|---|---|---|---|
| $p_1 > 0.5$, $p_2 + p_3 + p_4 < 0.5$ | $+$ | $-$ | $+$ | $\mathcal{E}_2$ |
| $p_2 > 0.5$, $p_1 + p_3 + p_4 < 0.5$ | $+$ | $+$ | $-$ | $\mathcal{E}_1$ |
| $p_3 > 0.5$, $p_1 + p_2 + p_4 < 0.5$ | $-$ | $-$ | $-$ | $\mathcal{E}_1$ |
| $p_4 > 0.5$, $p_1 + p_2 + p_3 < 0.5$ | $-$ | $+$ | $-$ | $\mathcal{E}_2$ |

When $\rho_{\text{BDS}}$ is entangled, the index for which $p_i > 1/2$ determines the sign of $a, b$, and $c$, see Table 3. It is notable that the signs of $a, b$ and $c$ follow a similar pattern to the Pauli basis expansion of various Depolarized Bell states listed in Table 2. We observe that, for suitable combinations of $a, b$, and $c \in \{+1, -1\}$, the Bell diagonal state reduces to one of the Depolarized Bell states, and states can be detected using the same WBMs, as in Table 2. Specifically, the value of $S$ under the two WBMs in Table 3 is equal to $(1-p_1-p_4)^2 - 4(p_1-p_4)^2$ and $(1-p_2-p_3)^2 - 4(p_2-p_3)^2$, respectively. Depending on the probabilistic mixture, one of the two WBMs will conclusively result in $S < 0$.

### 4.3 Two-qubit Amplitude Damping on Depolarized Bell States

A qubit amplitude damping channel is a source of noise in superconducting circuit-based quantum computing and thus serves as a realistic channel model for simulating lossy processes in these systems. Mathematically, it can be obtained from an isometry $J$,

$$J : \mathcal{H}_a \mapsto \mathcal{H}_b \otimes \mathcal{H}_c; \quad J^\dagger J = I_a \tag{11}$$

where $\mathcal{H}_a$ denotes the Hilbert space for the channel's input, and $\mathcal{H}_b$ and $\mathcal{H}_c$ represent the Hilbert spaces for the direct and complementary channel outputs, respectively. An isometry of the form,

$$
\begin{aligned}
J_1 \left|0\right\rangle_a &= \left|0\right\rangle_b \left|1\right\rangle_c, \\
J_1 \left|1\right\rangle_a &= \sqrt{1-r} \left|1\right\rangle_b \left|1\right\rangle_c + \sqrt{r} \left|0\right\rangle_b \left|0\right\rangle_c,
\end{aligned} \tag{12}
$$

where $0 \leq r \leq 1$ defines a pair of channels, $\mathcal{B}(A) = \text{Tr}_c(JAJ^\dagger)$ and $\mathcal{C}(A) = \text{Tr}_b(JAJ^\dagger)$. Here, $\mathcal{B}$ is an amplitude damping channel with damping probability $r$ for the state $\left|1\right\rangle_a$ to decay to output state $\left|0\right\rangle_b$. The isometry $J_1 = K_0 \otimes \left|0\right\rangle + K_1 \otimes \left|1\right\rangle$ where $K_0$ and $K_1$ (Kraus) damping operators such that $K_0 = [0, \sqrt{r}; 0, 0]$ and $K_1 = [1, 0; 0, \sqrt{1-r}]$. For a single qubit represented by state $\rho$, the amplitude-damped output is given by,

$$\mathcal{B}(\rho) = K_0 \rho K_0^\dagger + K_1 \rho K_1^\dagger. \tag{13}$$

We can extend equation 13 for two-qubit states with damping probabilities $r$ and $q$ for the first and second qubit, respectively. Assuming that $r = q$, we consider Depolarized Bell states equation 9 with amplitude damping.

**Proposition 3** *For any damping probability $r > 0$, a Depolarized Bell state with amplitude damping can not be expressed as a Bell diagonal state equation 10.*

This fact can be readily demonstrated through a straightforward calculation. Consider the Isotropic state $\rho(w) = w \left|\Phi^+\right\rangle \left\langle\Phi^+\right| + (1-w)\frac{I}{4}$, which can be represented by the Bell diagonal state formed with probability distribution $(p_1, p_2, p_3, p_4) = ((3w+1)/4, (1-w)/4, (1-w)/4, (1-w)/4)$. In a Bell diagonal state, the diagonal elements corresponding to $\left|00\right\rangle \left\langle00\right|$ and $\left|11\right\rangle \left\langle11\right|$ are identical. In the case of an amplitude-damped Isotropic state, we observe that,

$$p_2 = p_3 = \frac{1-r}{4} \left(w - wr - r - 1\right).$$

However, obtaining closed-form expressions for $p_1$ and $p_4$ when $r > 0$ is cumbersome. Specifically, the values on the diagonal corresponding to $\left|00\right\rangle \left\langle00\right|$ and $\left|11\right\rangle \left\langle11\right|$ is given by $w(r^2 + 1)/2 - (w-1)4(r+1)^2/4$ and $w(r-1)^2/2 - (w-1)(r-1)^2/4$, respectively. These expressions are equal only when $r = 0$.

**Proposition 4** *For every $w \in [\frac{1}{3}, 1]$, there exists $\tilde{r} \subset [0,1]$ such that an amplitude-damped Depolarised Bell state becomes separable.*

The PPT criterion asserts that a two-qubit state is entangled if and only if its partial transpose contains at least one negative eigenvalue. For Bell states that are both amplitude damped and depolarized, we evaluate the eigenvalues and observe that one of them can exhibit either positive or negative values contingent upon the range of $r$. Detailed findings are presented in Table 4 and depicted graphically in Fig. 1a and Fig. 1b. Furthermore, the WBM for amplitude damped and Depolarized Bell states aligns with that of depolarized Bell states, as outlined in Table 2.

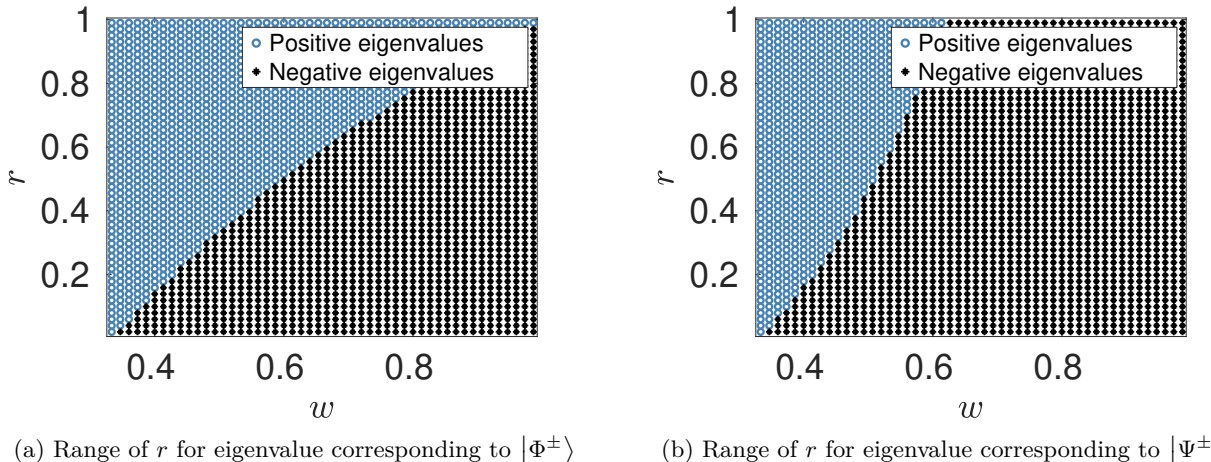

(a) Range of $r$ for eigenvalue corresponding to $\left|\Phi^{\pm}\right\rangle$      (b) Range of $r$ for eigenvalue corresponding to $\left|\Psi^{\pm}\right\rangle$

Figure 1: A phase diagram representing the region of damping and depolarizing parameters, $r$ and $w$, respectively, where the dampeddepolarized Bell state has negative or positive partial transpose.

Table 4: The four eigenvalues of amplitude-damped Depolarized Bell states

| State with $\left|\Phi\right\rangle^{\pm}$ | State with $\left|\Psi\right\rangle^{\pm}$ | **Sign of eigenvalue** |
|:---:|:---:|:---:|
| $\frac{(w+1)(1-r^2)}{4}$ | $\frac{(1-r)(1+r+w-wr)}{4}$ | Always positive |
| $\frac{(w+1)(1-r)^2}{4}$ | $\frac{(1-r)(1+r+w-wr)}{4}$ | Always positive |
| $\frac{w(r-1)^2+(r+1)^2}{4}$ | $\frac{r^2+1-w(1-r)^2+2\sqrt{w^2(1-r)^2+r^2}}{4}$ | Always positive |
| $\frac{-r^2(w-1)+wr+(1-3w)}{4}$ | $\frac{r^2+1-w(1-r)^2-2\sqrt{w^2(1-r)^2+r^2}}{4}$ | Positive and Negative |

## 5 Stochastic MAB policies for Entanglement Detection

We apply stochastic MAB algorithms for entanglement detection in the parameterized states $\mathcal{F}$ from Section 3. The terminology aligns with classical counterparts, as outlined in Table 1. Consider a set of $K$ unknown states, denoted by $\mathcal{A} = \{\rho_1, \rho_2, \ldots, \rho_K\} \in \mathcal{F}$. To perform measurements on the arms, the learner must know the underlying WBM. Thus, we assume knowledge of the specific forms of the arms in $\mathcal{A}$. For instance, $\mathcal{A}$ could represent the set of isotropic states detectable under the WBM $\mathcal{E}_2$, where each state is of the form $\rho_i = w_i \left|\Phi^+\right\rangle \left\langle\Phi^+\right| + (1-w_i)\frac{I}{4}$, with $w_i$ being unknown for all $i \in [K]$. With this assumption, we describe the MAB routine as follows: In each round $t \in \mathbb{N}$, the learner selects a state $\rho_{a_t} \in \mathcal{A}$, performs a measurement $\mathcal{E}$ and obtains i.i.d. outcomes, and then updates the values $\hat{\boldsymbol{S}}_{\mathcal{E}}$ to identify the entangled arm(s) or continues. In the subsequent sections, we discuss two MAB policies: *Successive Elimination*, which is applicable when there is a guarantee of one entangled arm among $K$ arms, and the *lil'HDoC* policy, designed for scenarios where there are $m$ entangled arms among $K$, with $m$ being unknown. For proving $\delta$-correctness of policies, we

use concentration inequalities applicable to $\sigma$-subgaussian[1] random variablesspecifically, the law of iterated logarithm (Jamieson et al., 2014, Lemma 3) for a finite sum of $\sigma$-subgaussian random variables:

**Lemma 5** *Let $X_1, X_2, \ldots X_t$ be i.i.d. sub-gaussian random variables with scale parameter $\sigma$. For any $\varepsilon \in (0,1)$, $\delta \in \left(0, \frac{\log(1+\varepsilon)}{e}\right)$, one has with probability at least $1 - c_\varepsilon \delta^{(1+\varepsilon)}$ for all $t \geq 1$,*

$$\frac{1}{t} \sum_{s=1}^{t} X_s \leq U(t, \delta), \tag{14}$$

*where $U(t, \delta) = (1 + \sqrt{\varepsilon})\sqrt{\frac{2\sigma^2(1+\varepsilon)}{t} \log\left(\frac{\log((1+\varepsilon)t)}{\delta}\right)}$ is the confidence width and $c_\varepsilon = \frac{2+\varepsilon}{\varepsilon}\left(\frac{1}{\log(1+\varepsilon)}\right)^{1+\varepsilon}$.*

### 5.1 Successive Elimination Algorithm

Consider the set of states $\mathcal{A} = \{\rho_1, \rho_2, \ldots, \rho_K\}$ detectable under WBM $\mathcal{E}$, with the guarantee that exactly one arm in the set is entangled. Given a WBM $\mathcal{E}$, for each state $\rho_i$, we use the notation $S_i$ and $S_\mathcal{E}(\rho_i)$ interchangeably, and denote by $\hat{S}_{i,N_i(t)}$ the estimate formed from $N_i(t)$ i.i.d. samples. The underlying problem instance $\mathbf{S}_\mathcal{E}$ satisfies $S_\mathcal{E}(\rho_1) \geq S_\mathcal{E}(\rho_2) \geq \cdots > S_\mathcal{E}(\rho_{K-1}) > 0 > S_\mathcal{E}(\rho_K)$. To identify the entangled arm, we use the Successive Elimination algorithm (Even-Dar et al., 2002) with a modified elimination rule, as outlined in Algorithm 1. The algorithm takes as input the set $\mathcal{A}$, threshold $\zeta = 0$, WBM $\mathcal{E}$ and the error probability $\delta$, and it outputs the entangled state $i^\star = \arg\min_{i \in [K]} S_\mathcal{E}(\rho_i)$. The algorithm maintains an active set $\Omega$ and measures every state in it. In order to identify $i^\star$, the policy eliminates states whose Lower Confidence Bound (LCB) exceeds the threshold and halts when only one state remains in the active set.

---

**Algorithm 1** Successive Elimination Algorithm

---

**Input:** $\zeta = 0$, $\delta$, $\mathcal{A}$, WBM $\mathcal{E}$
**Output:** $\Omega$
  Initialize active set $\Omega \leftarrow \mathcal{A}$
  Set: $t \leftarrow 0, N_i(t) \leftarrow 0, \hat{S}_{i,N_i(t)} \leftarrow 0, \ \forall i \in \Omega$
  **for** $t = 1, 2, 3, \ldots$ **do**
    **for** $\rho_i \in \Omega$ **do**
      Perform measurement $\mathcal{E}$ on $\rho_i$
      Update $N_i(t), \hat{S}_{i,N_i(t)}$ based on measurement outcomes
      Update confidence width $U\left(N_i(t), \frac{\delta}{c_\varepsilon K}\right)$ (see Lemma 5)
      Compute lower confidence bound: $\text{LCB}_i(t) \leftarrow \hat{S}_{i,N_i(t)} - U\left(N_i(t), \frac{\delta}{c_\varepsilon K}\right)$
    **end for**
    **if** $\text{LCB}_i(t) > 0$ for $i \in \Omega$ **then**
      Update active set: $\Omega \leftarrow \Omega - \{i\}$
    **end**
    **if** $|\Omega| = 1$ **then**
      Return $\Omega$
    **end**
  **end for**

---

**Lemma 6** *Algorithm 1 is $\delta$-correct.*

**Proof:** The proof is presented in Appendix A.2.1.     □

The correctness of Algorithm 1 and the copy complexity of identifying the entangled arm are presented below.

---

[1]A $\sigma$-subgaussian random variable is a real, centered random variable $X$ that satisfies $\mathbb{E}[e^{sX}] \leq e^{\sigma^2 s^2/2}$ for any $s \in \mathbb{R}$.

**Theorem 7** *With probability at least $1 - \delta$, the entangled state $i^\star = K = \arg\min_{i \in [K]} S_\mathcal{E}(\rho_i)$ remains in the active set $\Omega$ till termination.*

**Proof:**  The proof is presented in Appendix A.2.2.  □

**Theorem 8** *With probability at least $1 - \delta$, Algorithm 1 identifies the entangled state $i^\star$, requiring $\sum_{i \in [K]} \mathcal{O}\left(\Delta_i^{-2} \log\left(\frac{K \log \Delta_i^{-2}}{\delta}\right)\right)$ copies. Here, $\Delta_i = |S_\mathcal{E}(\rho_i) - \zeta|$ denotes the sub-optimality gap with respect to the threshold $\zeta$.*

**Proof:**  The proof is presented in Appendix A.2.3.  □

We observe that the sample complexity derived in Theorem 8 is within a $\log(K)$ factor of the optimal bound, as demonstrated in Theorem 1 of Jamieson et al. (2014). This result follows from the concentration bound established in Lemma 5, which forms the basis for the MAB policy described in the following section.

### 5.2 lil'HDoC Algorithm

The lil'HDoC algorithm (Tsai et al., 2024) builds on the HDoC algorithm (Kano et al., 2018) by leveraging finite LIL concentration bounds (Lemma 5) instead of the LCB-based identification rule (Kalyanakrishnan et al., 2012). To explore among promising arms, lil'HDoC adopts the sampling rule from Kano et al. (2018), derived from the UCB algorithm for regret minimization (Auer et al., 2002). It improves sample complexity over HDoC by utilizing the LIL bound, where the $\sqrt{\log \log t / t}$ factor has a higher growth rate than the $\sqrt{\log t / t}$ factor in the LCB bound. In other words, there exists a value $T$ such that for all $t > T$, $c_1, c_2 \in \mathbb{R}^+$,

$$c_1 \sqrt{\frac{\log t}{t}} > c_2 \sqrt{\frac{\log \log t}{t}}.$$

The confidence bound for HDoC grows as $\alpha(t) = \sqrt{\ln\left(\frac{4Kt^2}{\delta}\right)/2t}$. Through straightforward calculations, the smallest integer $T$ such that the confidence bound $U\left(T, \delta/c_\varepsilon K\right)$ is greater than $\alpha(T)$ is,

$$T \geq \frac{1}{4} \log(K + 1) \log\left(\max\left(\frac{1}{\delta}, 2\right)\right) c_\varepsilon^{3/2}. \tag{15}$$

Thus, if each state is measured $T$ times initially, lil'HDoC achieves comparable identification capabilities to HDoC with $\mathcal{O}\left(\log(K + 1) \log\left(\max\left(1/\delta, 2\right)\right)\right)$ copies of each state. We note that small values of $\epsilon$ tighten the confidence radius and therefore incur more samples before elimination, while large $\epsilon$ values reduce the number of samples with a higher chance of premature arm elimination. The asymptotic growth of $U(t, \delta)$ with $\epsilon$ is sublinear, and the policy's correctness remains unaffected for $\epsilon > 0$. The 'warm-start' phase parameter $T$ controls the number of measurements collected before adaptive allocation begins. Once the threshold in equation 15 is crossed, the $\delta$-correct guarantees and copy complexity depend primarily on $(\Delta_i, K, \delta)$ and not on $T$ itself.

Consider $K$ states such that $S_\mathcal{E}(\rho_1) \geq S_\mathcal{E}(\rho_2) \ldots > S_\mathcal{E}(\rho_{K-m}) > 0 > S_\mathcal{E}(\rho_{K-m+1}) \ldots > S_\mathcal{E}(\rho_K)$, with $m$ being unknown. The algorithm takes as input, the set of states $\mathcal{A}$, threshold $\zeta = 0$, WBM $\mathcal{E}$ and the error probability $\delta$ and outputs $\mathcal{A}_{\text{ent}} = \{i \in [K]$ such that $S_\mathcal{E}(\rho_i) < 0\}$. The algorithm maintains an active set $\Omega$ and terminates when the set $\Omega = \emptyset$. To demonstrate the correctness of Algorithm 2, we first show that the algorithm is $(\lambda, \delta)$-correct for all $\lambda \in [K]$ and then characterize the copy complexity of identifying $m$ entangled states.

**Lemma 9** *Algorithm 2 is $\delta$-correct.*

**Proof:**  *The proof is presented in Appendix A.3.1.*  □

**Theorem 10** *With probability at least $1 - \delta$, the algorithm identifies all the states in $\mathcal{A}_{ent}$.*

---

**Algorithm 2** LIL'HDOC ALGORITHM

---

**Input:** $\zeta = 0$, $\delta$, $\mathcal{A}$, WBM $\mathcal{E}$
**Output:** $\mathcal{A}_{\text{ent}}$

   Initialize active set $\Omega \leftarrow \mathcal{A}$, $\mathcal{A}_{\text{ent}} \leftarrow \emptyset$
   Set: $t \leftarrow 0, N_i(t) \leftarrow 0, \hat{S}_{i,N_i(t)} \leftarrow 0, \ \forall i \in \Omega$
   **for** $i = 1, 2, \ldots K$ **do**
      Perform $T$ measurements $\mathcal{E}$ on $\rho_i$
      Update $t, N_i(t), \hat{S}_{i,t}$ based on outcomes
   **end for**
   **while** $\Omega \neq \emptyset$ **do**
      Find $h_t = \arg\max_{i \in \mathcal{A}} \ \hat{S}_{i,N_i(t)} + \sqrt{\frac{\log t}{2N_i(t)}}$
      Perform measurement $\mathcal{E}$ on $\rho_{h_t}$
      Update $t, N_{h_t}(t), \hat{S}_{h_t, N_{h_t}(t)}$ based on outcomes
      Update confidence width $U\left(N_i(t), \frac{\delta}{c_\varepsilon K}\right)$
      **if** $\hat{S}_{h_t, N_{h_t}(t)} - U\left(N_{h_t}(t), \frac{\delta}{c_\varepsilon K}\right) \geq \zeta$ **then**
         Remove $h_t$ from $\Omega$
      **else if** $\hat{S}_{h_t, N_{h_t}(t)} + U\left(N_{h_t}(t), \frac{\delta}{c_\varepsilon K}\right) < \zeta$ **then**
         Add $h_t$ to $\mathcal{A}_{\text{ent}}$
         Remove $h_t$ from $\Omega$
      **end**
   **end while**

---

**Proof:** The proof is presented in Appendix A.3.2. $\qquad\square$

With $T = 1$ in equation 15, it can be seen from Theorem 8 that the number of samples required to identify an entangled state $\rho_i \in \mathcal{A}$ is $\mathcal{O}\left(\Delta_i^{-2} \log\left(\frac{K \log \Delta_i^{-2}}{\delta}\right)\right)$. However, in practice, $T$ is chosen to be larger than 1, and the total sample complexity is expressed in terms of $\Delta = \min_{i \in [K]} \Delta_i$.

**Theorem 11** *With probability $1 - \delta$ and an initialization of $T$ measurements, Algorithm 2 identifies the entangled states using $\mathcal{O}\left(\Delta^{-2}\left(K \log \frac{1}{\delta} + K \log K + K \log\log \frac{1}{\Delta}\right)\right) + \mathcal{O}\left(K \log(K+1) \log\left(\max\left(\frac{1}{\delta}, e\right)\right)\right)$ copies.*

**Proof:** The first term in the sample complexity is derived in Appendix A.2.3 and the second term follows from equation 15. $\qquad\square$

## 6 Implementation and Simulations on IBMQ Cloud

This section presents an experimental workflow for detecting entangled states from an ensemble of Bell Diagonal states. Sections 6.1 and 6.2 describe the procedures for generating Bell Diagonal states (BDS) and their corresponding WBMs, respectively. The performance of the MAB policies (see Section 5) is presented through numerical findings in Sections 6.4.

### 6.1 Generating Bell Diagonal States

Bell Diagonal States (BDS) are constructed as convex combinations of the four Bell states equation 10, forming a geometric tetrahedron, $\mathcal{T}$ and are represented by:

$$\rho_{\text{BDS}} = \sum_{j=1}^{4} p_j |\Upsilon\rangle \langle\Upsilon| = \frac{1}{4}\left[I + \sum_{j=1}^{3} t_j \sigma_j^A \otimes \sigma_j^B\right]. \tag{16}$$

Here, $\sigma_j$'s are the Pauli operators and $(t_1, t_2, t_3)$ are the coordinates within the tetrahedron $\mathcal{T}$. The mapping $\{p_j\}_{j=1}^4 \rightarrow (t_1, t_2, t_3)$ equation 17 is implemented through the quantum circuit proposed by Pozzobom & Maziero (2019); Riedel Gårding et al. (2021) and is shown in Fig. 2.

$$\sqrt{p_1} = \cos(\psi), \quad \sqrt{p_2} = \sin(\psi)\cos(\theta), \quad \sqrt{p_3} = \sin(\psi)\sin(\theta)\cos(\varphi), \quad \sqrt{p_4} = \sin(\psi)\sin(\theta)\sin(\varphi) \tag{17}$$

The sub-circuit $G$ encodes the probabilities $\{p_j\}_{j=1}^4$ into canonical coordinates $(\psi, \theta, \varphi)$ on the unit 3-sphere, and sub-circuit $B$ entangles the states in the Bell basis. Finally, the Bell-Diagonal state $\rho_{\text{BDS}} = \rho_{cd}$ is obtained by taking a partial trace over qubits $a$ and $b$.

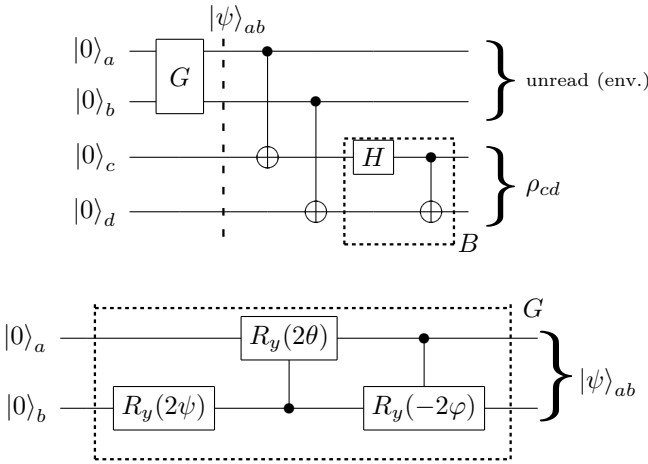

Figure 2: Four-qubit circuit for generating BDS with canonical encoder $G$ shown below.

## 6.2 Implementing Witness Basis Measurements

As outlined in Table 3, BDS are detectable under WBMs $\mathcal{E}_1$ and $\mathcal{E}_2$. To measure in the Pauli-Z (computational) basis, we apply appropriate unitary transformations to $\mathcal{E}_1$ and $\mathcal{E}_2$. The corresponding transformations are realized through circuits $\text{CIRC}_{\mathcal{E}_1}$ and $\text{CIRC}_{\mathcal{E}_2}$ shown in Fig. 3 and applied to qubits $c$ and $d$ (see Fig. 2) before measurement.

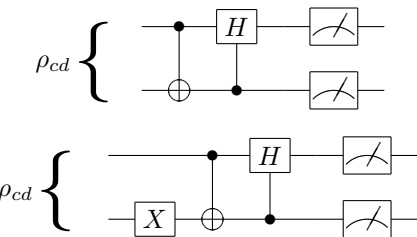

Figure 3: Circuits $\text{CIRC}_{\mathcal{E}_1}$ (top) and $\text{CIRC}_{\mathcal{E}_2}$ (bottom) perform the unitary transformations required to map $\mathcal{E}_1$ and $\mathcal{E}_2$ into the Pauli-Z basis.

## 6.3 Workflow for entanglement detection

We propose a workflow for detecting entanglement in BDS without assuming prior knowledge of the specific WBM. Instead, WBMs are sequentially adapted using suitable unitary transformations, as detailed in Section 2.1.2. To generate a set $\mathcal{A} = \{\rho_1, \rho_2, \ldots, \rho_K\}$ of BDS, we construct $K$ sets of probabilistic mixtures for combining the four Bell states. Specifically, $m$ states are generated with $\max_j p_j > \frac{1}{2}$, while the remaining $K - m$ states satisfy $\max_j p_j \leq \frac{1}{2}$. These probabilities are encoded following the procedure outlined in Fig. 2, where the BDS circuit for state $\rho_i$ is denoted as $\text{BDS}_i$. Subsequently, one of two WBM circuits, $\text{CIRC}_{\mathcal{E}_1}$ or

---

**Algorithm 3** Workflow for Entanglement Detection in BDS

---

**Input:** $\zeta = 0$, $\delta$, $\{\text{BDS}_i\}$, $\text{CIRC}_{\mathcal{E}}$, WBM choice $= 1$
**Output:** $A_{\text{ent}}$, Stopping time $\tau$
   Run Algorithm 2 on $\{\text{BDS}_i\}$ with circuit $\text{CIRC}_{\mathcal{E}_1}$ on $K$ states
   Return entangled states $|A_{\text{ent},1}| = \tilde{m}$ and stopping time $\tau_1$.
   **if** $(\tilde{m} = K)$ **then**
      $A_{\text{ent},2} \leftarrow \emptyset$, $\tau_2 \leftarrow 0$.
   **else if** $\tilde{m} < K$ **then**
      Run Algorithm 2 on $\{\text{BDS}_i\}$ with circuit $\text{CIRC}_{\mathcal{E}_2}$ on $K - \tilde{m}$ states
      Return entangled states $A_{\text{ent},2}$ and stopping time $\tau_2$.
   **end**
   $A_{\text{ent}} \leftarrow A_{\text{ent},1} + A_{\text{ent},2}$,   $\tau \leftarrow \tau_1 + \tau_2$

---

$\text{CIRC}_{\mathcal{E}_2}$, is appended to the respective BDS circuit. Algorithm 3 outlines this workflow for BDS and takes the following inputs: threshold $\zeta = 0$, error $\delta$, BDS circuits $\{\text{BDS}_i\}$, WBM circuits $\text{CIRC}_{\mathcal{E}_1}$ and $\text{CIRC}_{\mathcal{E}_2}$ and the initial choice of WBM. Notably, the initial WBM selection is arbitrary, as the sequence of WBM adaptations does not rely on state estimation.

The learner does not initially know under which WBM the BDS are detectable. Consequently, at least one iteration of Algorithm 2 must be executed. In the first iteration, the algorithm processes circuits corresponding to $K$ states with WBM $\mathcal{E}_1$ (or $\mathcal{E}_2$) and identifies a subset of entangled states, $\tilde{m}$, where $0 \leq \tilde{m} \leq K$. In the second iteration, Algorithm 2 is applied to the $K - \tilde{m}$ states that remain undetected by using circuits with WBM $\mathcal{E}_2$ (or $\mathcal{E}_1$) as inputs.

**Corollary 12** *Let* $\Delta_1 \coloneqq \min|\boldsymbol{S}_{\mathcal{E}_1}|$, $\Delta_2 \coloneqq \min|\boldsymbol{S}_{\mathcal{E}_2}|$ *and* $\Delta_{min} = \min\{\Delta_1, \Delta_2\}$, *then, with probability* $1 - \delta$ *and* $T = 1$, *Algorithm 3 identifies entangled BDS using* $\mathcal{O}\left(\Delta_{min}^{-2}\left(K\log\frac{1}{\delta} + K\log K + K\log\log\frac{1}{\Delta_{min}}\right)\right)$ *copies.*

### 6.4 Qiskit Experiment

The workflow presented in Algorithm 3 is simulated on IBM's Qiskit. The Python code implementation is available in Bharati (2025). We present numerical results on the achievable copy complexity for entanglement detection in BDS. The experimental setup is given as follows:

- **Simulation Environments:** The workflow is executed across three computational setups: (*i*) **AerSimulator** for idealized simulations, (*ii*) **FakeBrisbane** backend to simulate noisy quantum environments, and (*iii*) **ibm-brisbane** for real quantum processing unit (QPU) computations.

- **Problem Instance**: We consider $K = 5$ states of which $m = 3$ are entangled. The probabilities are suitably generated, and the true corresponding parameters under $\mathcal{E}_1$ and $\mathcal{E}_2$ are,

$$\boldsymbol{S}_{\mathcal{E}_1} = [0.6306, -0.2688, 0.5232, 0.1796, 0.0695],$$
$$\boldsymbol{S}_{\mathcal{E}_2} = [-0.0749, 0.5963, -0.1735, 0.2801, 0.3768].$$

- **MAB routine**: Each state was measured $10^6$ times on backends (*i, ii*) and $10^5$ times on (*iii*). Algorithm 3 was run 20 times on (*i, ii*) and 5 times on (*iii*) for $\delta \in (0, 1)$. We plot the average number of copies measured until stoppage on the y-axis and $\log(1/\delta)$ on the x-axis, as shown in Fig.4. Here, we note that the large standard deviation for the trend in backend (*iii*) arises due to the limited number of experiment iterations, constrained by available compute resources.

- **Benchmarking**: We benchmark our approach against non-adaptive BDS tomography, as described in Appendix A.1 and compare the resulting copy complexity across the same range of $\delta$. We present a criterion in equation 29 that guarantees that the reconstructed states are not misclassified (see

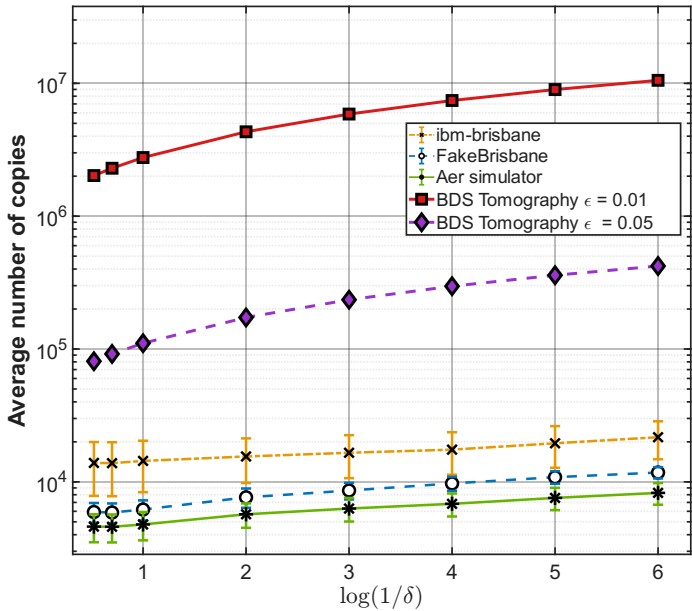

Figure 4: Copy complexity for entanglement detection in BDS v/s $\log(1/\delta)$

Appendix A.1). For the $K = 5$ BDS instance considered in this experiment, the bottleneck is determined by State 5 ($\max_i p_i = 0.446$), implying $\epsilon < 0.054$ to avoid misclassification theoretically. Choosing $\epsilon = 0.05$ allows only a negligible margin for statistical fluctuations, resulting in low classification accuracy. Thus, to ensure a valid comparison that matches the high reliability of the MAB approach, we choose $\epsilon = 0.01$ as the functional baseline.

From Corollary 12, we observe that the factor $\log(1/\delta)$ has a multiplicative effect on the sample complexity, while the average copy complexity is primarily determined by $\Delta_{\min}$. The values of $S_{\mathcal{E}}$ are governed by the four frequencies $f_1$, $f_2$, $f_3$, and $f_4$, as defined in equation 7. While the true values of the $f_i$'s are calculated using $\mathrm{Tr}\{\rho_{\mathrm{BDS}}E_i\}$, the values of $f_i$ obtained from register counts based on simulations performed on different backends differ from the true values upto $\mathcal{O}(10^{-2})$. Due to measurement noise and decoherence, the goalpost for $S_{\mathcal{E}}$ varies across different backends, and these differences influence $\Delta_{\min}$. One option is to mitigate the measurement noise (see details in the appendix, Sec. A.4).

## 7 Entanglement Detection in Arbitrary Quantum States

This section outlines a routine for detecting entanglement in arbitrary two-qubit quantum states. Specifically, we consider $K$ arbitrary states, one of which is entangled, and describe an MAB routine along with numerical results.

### 7.1 Numerical Experiment

The workflow outlined in Algorithm 4 is implemented in MATLAB. The algorithm takes the following inputs: a threshold $\zeta$, an error parameter $\delta$, a set of $K$ states $\mathcal{A}$ (with the promise that one state is entangled), and a permutation of $\{1, 2, 3, 4, 5, 6\}$ that specifies the order in which the WBMs should be adapted. As this is a promise problem, the algorithm terminates as soon as it identifies an entangled state, without needing to measure with all six WBMs. The different modules in the software code are described below:

- **Generating arbitrary quantum states**: To generate arbitrary density matrices, we follow the method described in Zyczkowski & Sommers (2001). Specifically, we start by generating a complex matrix $A \in \mathbb{C}^{4 \times 4}$, where the real and imaginary parts of each element are independently sampled

---

**Algorithm 4** Entanglement detection for arbitrary states

---

**Input:** $\zeta = 0$, $\delta$, $\mathcal{A} \leftarrow \{\rho_1, \rho_2 \ldots \rho_K\}$, WBM Order $P$
**Output:** $A_{\text{ent}}$
  flag $\leftarrow 1$, $I \leftarrow 1$
  **while** flag **do**
    With $\mathcal{E} \leftarrow \mathcal{E}_{P(I)}$, run Algorithm 2 for $K$ arms
    Return entangled arm $A_{\text{ent}}^{(I)}$
    **if** $|A_{\text{ent}}^{(I)}| = 1$ **then**
      flag $\leftarrow 0$
    **else**
      $I \leftarrow I + 1$
    **end**
  **end while**
  $A_{\text{ent}} \leftarrow A_{\text{ent}}^{(I)}$

---

from a normal distribution. We then compute the density matrix $\rho$ by normalizing $AA^\dagger$, resulting in $\rho = AA^\dagger/\text{Tr}(AA^\dagger)$. This procedure ensures that $\rho$ is a valid density matrix. We encountered pure states with $S_{\mathcal{E}}(\cdot) = 0$. For such cases, the algorithm took a significantly long time to converge and, despite this, incorrectly estimated the value of $S_{\mathcal{E}}(\rho)$. Consequently, we adjusted the threshold to $\zeta = -1 \times 10^{-3}$ and imposed a cutoff on the sample complexity at $1 \times 10^{12}$ to better reflect the real-time performance of this policy.

- **Experiment Setup**: In this experiment, we generate 1000 distinct instances of $K = 5$ full rank arbitrary states, ensuring that each instance contains exactly one entangled state. These instances are validated using the PPT criterion.

**Detection ratio versus $\delta$:** We test the efficacy of using the single-parameter family of witnesses equation 4 to detect entanglement in arbitrary states. For $\delta \in (0, 1)$, we report the *detection ratio*, which is defined as the fraction of times the entangled state is correctly identified by the MAB policy. This result is shown in Fig. 5. We observe that the detection ratio diminishes with larger error margins.

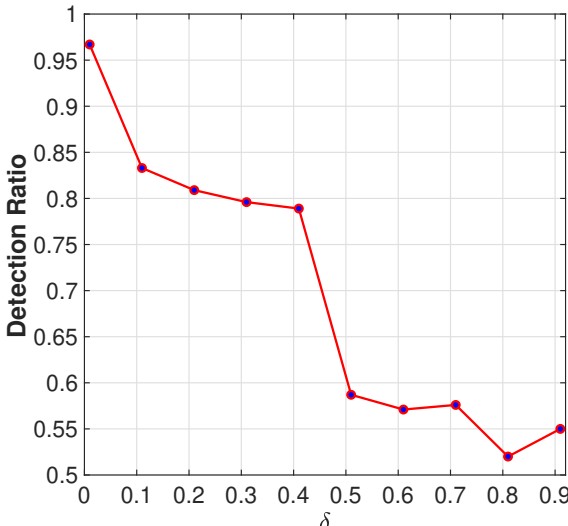

Figure 5: Entanglement Detection ratio v/s $\delta$ for arbitrary quantum states

**WBM usage under arbitrary ordering vs.** $\delta$**:**  For a random WBM ordering, we compute how many WBMs are required to detect a single valid entangled state among a set of $K$ states. For $\delta \in (0,1)$, we present the frequency distribution of the number of WBMs used, displayed as a cumulative histogram in Fig. 6. For significantly larger values of $\delta$, the lower detection ratios indicate that the algorithm terminates

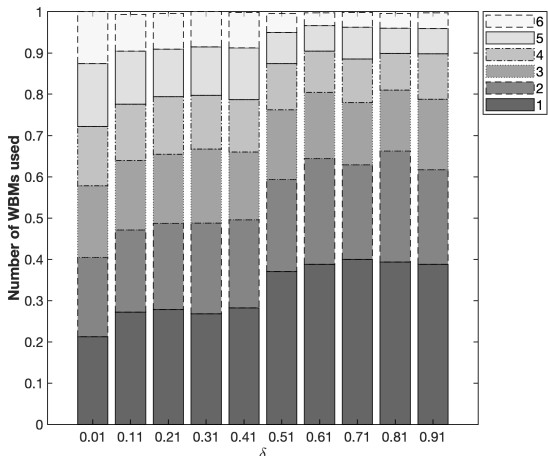

Figure 6: The cumulative histograms compare the number of WBMs used to detect one valid entangled state across different values of $\delta$.

upon identifying the wrong state, preventing further adaptation and primarily (around 85%) relying on up to three witnesses. This experiment can be extended to the scenario where there are $m$ entangled states. However, since $m$ is unknown and the states may be detectable under any of the WBMs, the routine would necessitate measuring under all WBMs to reliably identify the entangled states.

## 7.2 Numerical Outliers

We present an example of an entangled state certified by the PPT test that yields positive values for $S_{\mathcal{E}}(\rho)$ under all six WBMs. Consider the pure entangled states and their corresponding $\boldsymbol{S}_{\mathcal{E}}$ values, as shown in Table 5. The state $\rho = \sum_{i=1}^{3} p_i |\psi_i\rangle \langle\psi_i|$, where $|\psi_i\rangle$ are defined in Table 5, and $(p_i)_{i=1}^{3} = (0.2936, 0.0655, 0.6409)$, has a negative eigenvalue of $-0.029$ after applying the partial transpose, thus confirming it as an entangled state. However, the values of $(S_{\mathcal{E}}) = (0.0732, 0.1727, 0.1257, 0.1139, 0.0736, 0.0296)$ under the six WBMs are all non-negative. This indicates that the state cannot be detected by the witness family described in equation 4.

Table 5: Examples of arbitrary pure entangled states detected by the family of witnesses equation 4

| Pure entangled states $|\psi_1\rangle, |\psi_2\rangle$ and $|\psi_3\rangle$ | Values under $(S_{\mathcal{E}_i})_{i=1}^{6}$ |
|---|---|
| $[0.2687 + 0.0375i; 0.2406 + 0.4090i; 0.0502 + 0.6162i; 0.2413 + 0.5107i]$ | $(-0.1851, 0.3160, 0.1598, -0.0058, 0.2177, -0.1947)$ |
| $[0.0565 + 0.3355i; 0.0508 + 0.0686i; 0.4885 + 0.5191i; 0.5689 + 0.2125i]$ | $(0.1562, -0.0280, -0.1135, 0.1832, -0.0779, 0.1373)$ |
| $[0.1953 + 0.4438i; 0.4958 + 0.4009i; 0.0069 + 0.3495i; 0.0322 + 0.4848i]$ | $(-0.1851, 0.3160, 0.1598, -0.0058, 0.2177, -0.1947)$ |

We derive an observation on the nature of such states, focusing specifically on the eigenstate $|\lambda\rangle_{\max} = [0.3773 - 0.1445i, 0.4768 - 0.3244i, 0.4598 + 0.0809i, 0.5351]$, which corresponds to the largest eigenvalue of $\rho$. This eigenstate has a Schmidt coefficient close to, but not exactly equal to 1, suggesting that it lies near the boundary of separable states while still remaining entangled. The pure state $|\lambda\rangle_{\max} \langle\lambda|_{\max}$ produces the following values for $(S_{\mathcal{E}}) = (0.0380, 0.1269, 0.0401, 0.1054, 0.0221, 0.0074)$. This highlights that both pure and mixed entangled states can yield inconclusive results when measured using this specific witness family. In these cases, it is crucial to measure all six witnesses a sufficient number of times to accurately obtain

the expected values of the corresponding observables. Additionally, performing FST can help determine the entanglement of these states using other separability criteria.

# 8 Discussions

## 8.1 The MAB Advantage

While traditional FST and fixed witness testing methods (Zhu et al., 2010) are direct and computationally feasible for bipartite qubit states, the proposed MAB approach provides a practical and theoretical advantage for entanglement detection in parameterised two-qubit states considered in this work.

**Fixed-confidence guarantees without state reconstruction:**  $\delta$-correct MAB policies considered in this work output a statistically certified set of entangled parameterised two-qubit states, obtained directly from measurement data and without explicit state reconstruction. In contrast, FST necessitates state reconstruction up to a specified trace accuracy $\epsilon$ and does not provide any explicit confidence guarantee on the entanglement aspect. Thus, any guarantee derived from FST is therefore indirect and depends on post-hoc analysis of the reconstructed state using analytical separability tests, whereas the MAB approach delivers decision-level guarantees by design.

**Adaptive allocation of measurements:**  The key advantage of the MAB formulation is its *adaptivity* in measuring the witness basis on states based on their separability gaps $\{\Delta_i\}$, i.e., prioritising states whose gaps are smaller or whose entanglement status is uncertain. This yields instance-dependent copy complexity guarantees that scale polynomially in $\Delta_i$ and have polylogarithmic dependence on $K$ and $(1/\delta)$. Adaptivity also allows early stopping when the criteria for entanglement certification are met. In contrast, *non-adaptive* methods like FST and repeated witness-testing approaches (Zhu et al., 2010) tend to systematically over-sample with no principled stopping rule; that is, states far from the threshold (large $\Delta_i$) are measured as extensively as the ones close to it (small $\Delta_i$) till the desired trace accuracy is achieved. This results in the copy complexity scaling linearly in $K$; sample-optimal FST with collective measurements requires $\mathcal{O}(16K/\epsilon^2)$ copies Haah et al. (2017).

**Practical Advantage:**  As illustrated in Fig. 4, the copy complexity achieved by the MAB policy is benchmarked against that of the non-adaptive tomography baseline for Bell-diagonal states described in Appendix A.1. The MAB routine is run across IBMQ backends (Aer, FakeBrisbane, ibm-brisbane), achieving a copy complexity of $\mathcal{O}(10^4) - \mathcal{O}(10^5)$. In contrast, the Bell Diagonal state tomography approach requires $\mathcal{O}(10^6) - \mathcal{O}(10^7)$ copies to achieve a trace distance accuracy $\epsilon = 0.01$ for the same scale ($K = 5$) and same range of $\delta$. We emphasise that the two-order-of-magnitude reduction in copy complexity is enabled via adaptive sampling and early stopping.

Overall, the MAB-based approach offers a quantitatively demonstrated reduction in copy complexity for Bell Diagonal states, explicit confidence guarantees, adaptivity in distributing measurement effort and scalability for batch detection tasks for the two-qubit parameterised states.

## 8.2 Does the MAB routine optimize WBM ordering?

As outlined in Dai et al. (2014), there are WBM optimization strategies that prescribe an optimal WBM ordering for efficiently detecting whether a *single* arbitrary two-qubit quantum state is entangled. One such adaptive strategy uses the maximum-likelihood maximum-entropy (MLME) estimate of the unknown state, based on causal measurement data. Using this estimate, the subsequent WBM is identified to be the one that minimises the quadratic separability criterion. This leads to partial estimation of the quantum state.

In the context of batch entanglement detection, where an unknown number $m$ of entangled states out of a set of $K$ states may be detectable under different witnesses, implementing the WBM adaptive strategies from Dai et al. (2014) would be both time-consuming and complex. This is because each of the $K$ states may require a unique permutation of the WBM ordering. Furthermore, the goal of the proposed MAB framework is to *minimize* the number of measurements needed for detecting entanglement in a given set of quantum

states under a specific WBM. Notably, this framework *does not* optimize the WBM ordering across multiple MAB runs.

The closest comparison is with Fig. 6, which depicts the cumulative frequency of WBMs used. This aligns with the cumulative percentage of states identified under the WBM family, as seen in schemes 1A and 4A of the recently reported incomplete state estimation techniques (see (Dai et al., 2014, Fig. 1)). However, this approach does not address the batch entanglement detection problem. The WBM adaptation scheme A in Dai et al. (2014) successfully detects 98% of random pure states but only 33% of full-rank mixed states. We specifically analyze the latter category, generating multiple instances of $K$ states to quantify the number of WBMs required to detect a single entangled state, presenting results for varying $\delta$. In this way, we provide numerical insight into the sample complexity and convergence rate of our proposed schema, in contrast to Dai et al. (2014).

## 9  Future Work And Conclusion

Batch entanglement detection, as discussed in this paper, is particularly useful for verifying the integrity of a batch of practically relevant entangled states, before use in applications like secure multi-channel quantum communication. We established a novel correspondence between the problem of batch entanglement detection and the Thresholding Bandit problem in stochastic Multi-Armed Bandits. We proposed the $(m, K)$-quantum Multi-Armed Bandit framework for entanglement detection. The focus of this framework is on identifying $m$ entangled states out of $K$ states, where $m$ is potentially unknown. We apply this framework to two-qubit states using two key ingredients: a specialized set of six measurements for two-qubit states called Witness Basis Measurements (WBM) $\mathcal{E}$ and a separability criterion $S_{\mathcal{E}}$, which is based on the data obtained from these measurements and serves as the parameter that needs to be estimated. We present theoretical guarantees and numerical simulations to demonstrate how this parameter can be estimated quickly and accurately using MAB policies. First, we show that entangled states belonging to a class of parameterised two-qubit states $\mathcal{F}$ can be detected by measuring a subset of the six WBMs. With the knowledge of the WBM, we show that we can directly apply some suitable MAB policies. Second, for the same parameterised states, we present a routine for entanglement detection when the WBM is not known by enabling arbitrary sequential adaptation of the WBMs. We extend this to arbitrary two-qubit quantum states and provide numerical results on the efficacy of using these measurements for detecting entanglement.

An exciting avenue for future research lies in identifying WBMs for higher-dimensional bipartite systems. The minimalistic tomographic scheme proposed in Zhu et al. (2010) significantly reduces the number of required witnesses for two-qutrits from 81 to just 11, demonstrating the potential for more efficient entanglement detection. Meanwhile, recent advancements in data-driven machine learning, particularly the use of SVMs to construct linear entanglement witnesses from local measurements (Greenwood et al., 2023), open new possibilities for tackling the $(m, K)$-quantum MAB problem. By leveraging these techniques, one could optimize the number of witnesses needed to reliably detect all $m$ states.

Entanglement detection can be reframed as a membership problem, where a state belongs to a set if it exhibits a specific propertysuch as entanglement. This perspective aligns with the partition identification problem (Juneja & Krishnasamy, 2019), in which the objective is to determine the partition to which a data point belongs using a hyperplane structure. Extending this framework to the $(m, K)$-quantum MAB problem could pave the way for groundbreaking approaches to adaptive entanglement detection.

## Acknowledgement

K.B. sincerely acknowledges the support from the Ministry of Education, Government of India, through the Prime Minister's Research Fellowship (PMRF) Scheme. V.S. is supported by the U.S. Department of Energy, Office of Science, National Quantum Information Science Research Centers, Co-design Center for Quantum Advantage (C2QA) contract (DE- SC0012704). K.J. gratefully acknowledges a grant from Mphasis to the Centre for Quantum Information, Communication, and Computing (CQuICC) at IIT Madras.

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

# A Appendix

## A.1 Non-adaptive baseline: Bell-Diagonal State Tomography

**Copy Complexity Bound:** For a prescribed trace-distance accuracy $\epsilon$, we derive a copy-complexity guarantee for non-adaptive Bell-diagonal-state tomography and compare it with adaptive bandit algorithms for entanglement detection. As seen in Table 3, Bell-diagonal states are characterised by three two-qubit Pauli correlations $XX, YY$ and $ZZ$. Thus, no other Pauli settings are required. Given $\rho_{\text{BDS}}$, it is sufficient to estimate

$$c_{xx} = \langle XX \rangle_{\rho_{\text{BDS}}}, \quad c_{yy} = \langle YY \rangle_{\rho_{\text{BDS}}}, \quad c_{zz} = \langle ZZ \rangle_{\rho_{\text{BDS}}}. \tag{18}$$

Let $\mathbf{c} := (c_{xx}, c_{yy}, c_{zz})^\top$. The eigenvalues of $\rho_{\text{BDS}}$ correspond to the statistical mixture $\mathbf{p} = \{p_i\}_i$ equation 10 and are given by the affine map

$$\mathbf{p} = \frac{1}{4} \left( \mathbf{1} + \mathbf{A}\mathbf{c} \right), \tag{19}$$

where

$$\mathbf{A} = \begin{pmatrix} 1 & -1 & 1 \\ 1 & 1 & -1 \\ -1 & -1 & -1 \\ -1 & 1 & 1 \end{pmatrix}.$$

Each measurement outcome for $s \in \{xx, yy, zz\}$ is a random variable $X_s \in \{-1, 1\}$ with mean $c_s$. Let $\hat{c}_s := 1/t \sum_j X_{s,j}$ be the empirical mean obtained from $t$ measurements. By Hoeffding's inequality,

$$\mathbb{P}\big(|\hat{c}_s - c_s| \geq \eta\big) \leq 2 \exp\left(-\frac{t\eta^2}{2}\right).$$

Denoting $\hat{\mathbf{c}} := (\hat{c}_{xx}, \hat{c}_{yy}, \hat{c}_{zz})^\top$ and taking a union bound over the three correlators gives,

$$\mathbb{P}\big(\|\hat{\mathbf{c}} - \mathbf{c}\|_\infty > \eta\big) \leq \sum_s \mathbb{P}\big(|\hat{c}_s - c_s| > \eta\big) \ \leq \ 6\exp\left(-\frac{t\eta^2}{2}\right) = \delta. \tag{20}$$

From equation 19, $\hat{\mathbf{p}} - \mathbf{p} = \frac{1}{4}A(\hat{\mathbf{c}} - \mathbf{c})$, hence

$$\|\hat{\mathbf{p}} - \mathbf{p}\|_1 = \tfrac{1}{4}\|A(\hat{\mathbf{c}} - \mathbf{c})\|_1 = \tfrac{1}{4}\sum_{i=1}^{4}\big|(A(\hat{\mathbf{c}} - \mathbf{c}))_i\big| \leq 3\|\hat{\mathbf{c}} - \mathbf{c}\|_\infty. \tag{21}$$

Let $\hat{\rho}_{\text{BDS}}$ be the estimate of $\rho_{\text{BDS}}$, then the trace distance between $\hat{\rho}_{\text{BDS}}$ and $\rho_{\text{BDS}}$ is given by,

$$D(\hat{\rho}_{\text{BDS}}, \rho_{\text{BDS}}) \ = \ \tfrac{1}{2}\|\hat{\rho}_{\text{BDS}} - \rho_{\text{BDS}}\|_1 \ \overset{(a)}{=} \ \tfrac{1}{2}\sum_{i=1}^{4}|\hat{p}_i - p_i| \ = \ \tfrac{1}{2}\|\hat{\mathbf{p}} - \mathbf{p}\|_1 \ \leq \ \tfrac{3}{2}\|\hat{\mathbf{c}} - \mathbf{c}\|_\infty \tag{22}$$

where (a) holds because both $\rho_{\text{BDS}}$ and $\hat{\rho}_{\text{BDS}}$ are diagonal in the Bell basis, so the eigenvalues of $\hat{\rho}_{\text{BDS}} - \rho_{\text{BDS}}$ are $\{\hat{p}_i - p_i\}_{i=1}^4$. To guarantee a trace distance of $\epsilon$ between $\hat{\rho}_{\text{BDS}}$ and $\rho_{\text{BDS}}$, it suffices to enforce $\|\hat{\mathbf{c}} - \mathbf{c}\|_\infty \leq \frac{2}{3}\epsilon$. Setting $\eta = \frac{2}{3}\epsilon$, we solve for $t$ in equation 20,

$$t \ \geq \ \frac{9}{2\epsilon^2}\log\left(\frac{6}{\delta}\right) \tag{23}$$

shots per setting. Since three measurement settings are used, the total number of copies required is,

$$N_{\text{BDS}}(\epsilon, \delta) \ = \ 3t \ \geq \ \frac{27}{2\epsilon^2}\log\left(\frac{6}{\delta}\right). \tag{24}$$

**Choice of $\epsilon$ for preserving the true status of reconstructed Bell-Diagonal States:** From Section 4.2, we know that a Bell diagonal state is entangled if $p_{i^\star} > \frac{1}{2}$, where $i^\star = \arg\max_i p_i$ and separable if $p_{i^\star} < \frac{1}{2}$. Since $\mathbf{p}$ and $\hat{\mathbf{p}}$ both form valid probability distributions,

$$\sum_{i=1}^{4}(\hat{p}_i - p_i) = 0. \tag{25}$$

Preserving the status of $\hat{\rho}_{\text{BDS}}$ amounts to preserving the inequality defining the status, i.e., entangled means $\hat{p}_{i^\star} > \frac{1}{2}$ and separable means $\hat{p}_{i^\star} < \frac{1}{2}$. Let $x := |\hat{p}_{i^\star} - p_{i^\star}|$ denote the magnitude of the estimation error on the largest eigenvalue. We consider the worst case in which $\hat{p}_{i^\star}$ changes by $x$ in the most detrimental direction to status preservation. If $\hat{p}_{i^\star}$ decreases by $x$, then

$$\hat{p}_{i^\star} - p_{i^\star} = -x.$$

For equation 25 to hold, we must have

$$\sum_{i \neq i^\star}(\hat{p}_i - p_i) = x.$$

By the triangle inequality,

$$\sum_{i \neq i^\star}|\hat{p}_i - p_i| \ \geq \ \left|\sum_{i \neq i^\star}(\hat{p}_i - p_i)\right| = x.$$

Substituting in equation 22,

$$D(\hat{\rho}_{\text{BDS}}, \rho_{\text{BDS}}) = \frac{1}{2}\left(|\hat{p}_{i^\star} - p_{i^\star}| + \sum_{i \neq i^\star}|\hat{p}_i - \hat{p}_i|\right) \geq x$$

Imposing the constraint $D(\hat{\rho}_{\text{BDS}}, \rho_{\text{BDS}}) \leq \epsilon$ therefore yields

$$|\hat{p}_{i^\star} - p_{i^\star}| \leq \epsilon. \tag{26}$$

We now present the condition for status preservation. If $\rho_{\text{BDS}}$ is entangled, then $p_{i^\star} > \frac{1}{2}$. The worst-case decrease according to equation 26 implies $\hat{p}_{i^\star} \geq p_{i^\star} - \epsilon$. Thus, entanglement is preserved if,

$$\epsilon < p_{i^\star} - \tfrac{1}{2}. \tag{27}$$

Similarly, if $\rho_{\text{BDS}}$ is separable, then $p_{i^\star} \leq \frac{1}{2}$, and the worst-case increase is $\hat{p}_{i^\star} \leq p_{i^\star} + \epsilon$. Hence, separability is preserved if

$$\epsilon < \tfrac{1}{2} - p_{i^\star}. \tag{28}$$

Combining equation 27 and equation 28, the unified status-preservation condition is,

$$\epsilon < \left| p_{i^\star} - \tfrac{1}{2} \right|. \tag{29}$$

### A.2 Proofs for Section 5.1

The following lemma is useful for some calculations.

**Lemma 13** *For $t \geq 1, c > 0, \varepsilon \in (0,1), 0 < w \leq 1$,*

$$\frac{1}{t} \log\left( \frac{\log\left((1+\varepsilon)t\right)}{w} \right) \geq c \implies t \leq \frac{1}{c} \log\left( \frac{2 \log\left( \frac{(1+\varepsilon)}{cw} \right)}{w} \right). \tag{30}$$

#### A.2.1 Proof of Lemma 6

**Proof:** Let $\mathcal{B}$ denote the "good" event that at any time $t > 0$ and for all arms $i \in [K]$, the true value $S_{\mathcal{E}}(\rho_i)$ is well concentrated around its estimate $\hat{S}_{i,N_i(t)} = 1/N_i(t) \sum_{s=1}^{t} J_{i,s}$, where i.i.d. samples $J_{i,s} := 4\mathbf{1}_{\{Y=1\}} \mathbf{1}_{\{Y'=2\}} - \left( \mathbf{1}_{\{Y=3\}} - \mathbf{1}_{\{Y=4\}} \right) \left( \mathbf{1}_{\{Y'=3\}} - \mathbf{1}_{\{Y'=4\}} \right)$ for i.i.d. outcomes $Y, Y'$ from measuring $\mathcal{E}$ on $\rho_i$.

$$\mathcal{B} := \bigcup_{i=1}^{K} \bigcup_{t=1}^{\infty} \left\{ |\hat{S}_{i,N_i(t)} - S_i| \leq U\left( N_i(t), \frac{\delta}{c_\varepsilon K} \right) \right\}$$

From Lemma 5 and by applying the union bound, we get that

$$\mathbb{P}\left[ \mathcal{B} \right] \geq 1 - c_\varepsilon K \left( \frac{\delta}{c_\varepsilon K} \right)^{1+\varepsilon} \geq 1 - \delta \tag{31}$$

where Eq. 31 holds because $\varepsilon \in (0,1)$ and $c_\varepsilon \geq 1$. $\qquad \square$

#### A.2.2 Proof of Theorem 7

**Proof:** Recall that the threshold $\zeta = 0$ and problem instance $\boldsymbol{S_{\mathcal{E}}}$ is such that $S_{\mathcal{E}}(\rho_1) \geq S_{\mathcal{E}}(\rho_2) \geq S_{\mathcal{E}}(\rho_3) \ldots > S_{\mathcal{E}}(\rho_{K-1}) > 0 > S_{\mathcal{E}}(\rho_K)$. Let us consider the case that the event $\mathcal{B}$ described in Lemma 6 holds. As outlined in Algorithm 1, the arm $i^\star$ will be dropped from the active set $\Omega$ if $\text{LCB}_{i^\star}(t) > 0$. That is,

$$\hat{S}_{i^\star, N_{i^\star}(t)} - U\left( N_{i^\star}(t), \frac{\delta}{c_\varepsilon K} \right) > 0$$
$$\hat{S}_{i^\star, N_{i^\star}(t)} - |\hat{S}_{i^\star, N_i^\star(t)} - S_{i^\star}| > 0$$
$$\implies S_{i^\star} > 0$$

This contradicts the assumption about the problem instance $\boldsymbol{S}$ because $S_{i^\star} = S_{\mathcal{E}}(\rho_K) < 0$ and so, the arm $i^\star$ will not be dropped from the active set $\Omega$ as long as event $\mathcal{B}$ holds. $\qquad \square$

### A.2.3 Proof of Theorem 8

**Proof:** Let us consider the case where $\mathcal{B}$ holds. By the elimination rule of Algorithm 1, an arm $i$ is removed from the active set $\Omega$ if $\text{LCB}_i(t) > 0$. We have that,

$$\hat{S}_{i,N_i(t)} - U\left(N_i(t), \frac{\delta}{c_\varepsilon K}\right) \geq \zeta$$

$$\hat{S}_{i,N_i(t)} - S_i + \Delta_i \geq U\left(N_i(t), \frac{\delta}{c_\varepsilon K}\right)$$

$$\implies \Delta_i \geq 2U\left(N_i(t), \frac{\delta}{c_\varepsilon K}\right) \tag{32}$$

Let us denote $N_i$ to be the number of samples of arm $i$, that is, $N_i = \inf\{t : U\left(N_i(t), \frac{\delta}{c_\varepsilon K}\right) \leq \frac{\Delta_i}{2}\}$. The minimum value of $N_i$ can be obtained by solving,

$$U\left(N_i, \frac{\delta}{c_\varepsilon K}\right) = \frac{\Delta_i}{2}$$

$$(1+\sqrt{\varepsilon})\sqrt{\frac{2(1+\varepsilon)}{N_i}\log\left(\frac{\log\left((1+\varepsilon)N_i\right)}{\delta/c_\varepsilon K}\right)} = \frac{\Delta_i}{2}$$

$$\frac{1}{N_i}\log\left(\frac{\log\left((1+\varepsilon)N_i\right)}{\delta/c_\varepsilon K}\right) = \frac{\Delta_i^2}{8(1+\varepsilon)(1+\sqrt{\varepsilon})^2} \tag{33}$$

From Lemma 13, we get that,

$$N_i = \frac{8(1+\varepsilon)(1+\sqrt{\varepsilon})^2}{\Delta_i^2}\log\left(\frac{2c_\varepsilon K\log\left(\frac{8c_\varepsilon(1+\varepsilon)^2(1+\sqrt{\varepsilon})^2}{\delta}\frac{K}{\Delta_i^2}\right)}{\delta}\right) \tag{34}$$

Thus, the total number of samples required to identify the arm $i^\star$ with a probability of at least $1 - \delta$ is $N \leq \sum_{i=1}^K N_i$. $\qquad\square$

### A.3 Proofs for Section 5.2

### A.3.1 Proof of Lemma 9

**Proof:** Firstly, we show that Algorithm 2 is $(\lambda, \delta)$-correct for arbitrary $\lambda \in [K]$. In the case where there are arms greater than or equal to $\lambda$, we show that $\mathbb{P}\left[\{\hat{m} < \lambda\} \cup \bigcup_{i \in \mathcal{A}_{\text{ent}}}\{S_i < \zeta\}\right] \leq \delta$ where $\hat{m}$ is the number of good arms identified by the agent. Since we are now considering the case when $m \geq \lambda$, the event $\{\hat{m} < \lambda\}$ implies that at least one good arm $j \in [m]$ is identified as a bad arm by the agent. That is, for some $j \in [m]$ and $t \in \mathbb{N}$, the upper confidence bound $\hat{S}_{j,N_j(t)} + U\left(N_j(t), \frac{\delta}{c_\varepsilon K}\right) < \zeta$. Thus, we have that,

$$\mathbb{P}[\hat{m} < \lambda] \leq \sum_{j \in [m]}\mathbb{P}\left[\bigcup_{t \in \mathbb{N}}\{\hat{S}_{j,N_j(t)} + U\left(N_j(t), \frac{\delta}{c_\varepsilon K}\right) < \zeta\}\right]$$

$$\leq \sum_{j \in [m]} c_\varepsilon\left(\frac{\delta}{c_\varepsilon K}\right)^{1+\varepsilon} \qquad \text{(By Lemma 5)}$$

$$\leq mc_\varepsilon\left(\frac{\delta}{c_\varepsilon K}\right) \tag{35}$$

The event $\bigcup_{i \in \{\hat{X}_1, \hat{X}_2, \dots \hat{X}_\lambda\}}\{\mu_i < \zeta\}$ considers all those outcomes where a bad arm is identified to be a good one. Thus, for some bad arm $j \in \{\hat{X}_1, \hat{X}_2, \dots \hat{X}_{\hat{m}}\}$ such that $j \in [K] \setminus [m]$, we have,

$$\mathbb{P}\left[\bigcup_{i\in\{\hat{X}_1,\hat{X}_2,\dots\hat{X}_\lambda\}}\{S_i < \zeta\}\right]$$

$$\leq \sum_{j\in[K]\setminus[m]}\mathbb{P}\left[\bigcup_{t\in\mathbb{N}}\{\hat{S}_{j,N_j(t)} - U\left(N_j(t),\frac{\delta}{c_\varepsilon K}\right) > \zeta\}\right]$$

$$\leq (K-m)c_\varepsilon\left(\frac{\delta}{c_\varepsilon K}\right) \tag{36}$$

Thus, putting Eq. 35 and Eq. 36 together, we get that $\mathbb{P}\left[\{\hat{m} < \lambda\} \cup \bigcup_{i\in\{\hat{X}_1,\hat{X}_2,\dots\hat{X}_{\hat{m}}\}}\{\mu_i < \zeta\}\right] \leq \delta$. Next, we consider the case when the number of good arms $m$ is less than $\lambda$ and show that $\mathbb{P}[\hat{m} \geq \lambda] \leq \delta$. Since there are at most $\lambda$ good arms, the event $\{\hat{m} > \lambda\}$ implies that one of the output arms $j \in \{\hat{X}_1, \hat{X}_2, \dots \hat{X}_\lambda\}$ is such that there exists some index $j$ such that $\hat{X}_j$ is a bad arm. Thus, we have that,

$$\mathbb{P}[\hat{m} \geq \lambda] \leq \sum_{j\in[K]\setminus[m]}\mathbb{P}[j \in \{\hat{X}_1, \hat{X}_2, \dots \hat{X}_\lambda\}]$$

$$\leq (K-m)c_\varepsilon\left(\frac{\delta}{c_\varepsilon K}\right)^{1+\varepsilon}$$

$$\leq \frac{K-m}{K}c_\varepsilon\left(\frac{\delta}{c_\varepsilon}\right)$$

$$\leq \delta \tag{37}$$

We see that the algorithm is $(\lambda, \delta)$-correct for all such $\lambda \in [K]$, thereby giving us that the algorithm is $\delta$-correct. $\square$

### A.3.2 Proof of Theorem 10

**Proof:** Recall that the threshold $\zeta = 0$ and problem instance $\boldsymbol{S}_\mathcal{E}$ is such that $S_\mathcal{E}(\rho_1) \geq S_\mathcal{E}(\rho_2)\dots > S_\mathcal{E}(\rho_{K-m}) > 0 > S_\mathcal{E}(\rho_{K-m+1})\dots > S_\mathcal{E}(\rho_K)$, with $m$ being unknown. Let us consider the case that the event $\mathcal{B}$ described in Lemma 6 holds. As outlined in Algorithm 2, an arm $i$ will be dropped if $\text{LCB}_i(t) > 0$. That is,

$$\hat{S}_{i,N_i(t)} - U\left(N_i(t),\frac{\delta}{c_\varepsilon K}\right) > 0$$

$$\hat{S}_{i,N_i(t)} - |\hat{S}_{i,N_i(t)} - S_i| > 0$$

$$\implies S_i > 0$$

Thus, as long as event $\mathcal{B}$ holds, all the arms that have $S_\mathcal{E} < 0$ will not dropped. Thus the lil'HDoC algorithm identifies all the arms correctly. $\square$

### A.4 Integrating Error Mitigation in MAB Algorithms for Batch Entanglement Detection

In the MAB-based workflow for entanglement detection described in Section 6, one state is measured at every time instant as dictated by the sampling rule, and the statisticsnamely, the estimates of $f_1, f_2, f_3$, and $f_4$are updated as new measurement outcomes are obtained. These estimates are susceptible to measurement errors, particularly readout errors, which induce inaccuracies in the measurement counts. To improve the accuracy of the estimates, we characterize such errors and wish to mitigate them (Qiskit Community, 2024). To this end, we carry out a preliminary investigation by incorporating a procedure for (a) error mitigation and (b) including error mitigation in the MAB routine, and study the impact of error mitigation on the overall copy complexity of batch entanglement detection.

### A.4.1 Procedure for Error Mitigation

In Fig. 3, we apply a unitary transformation to WBM $\mathcal{E}_1$ and $\mathcal{E}_2$ to measure the state of the system $\rho$ in the computational (Pauli Z) basis. Consequently, we obtain expectation values of the diagonal Pauli operators $ZZ$, $ZI$, and $IZ$. The estimates of $f_1$, $f_2$, $f_3$, and $f_4$ are linear combinations of these expectation values.

$$
\begin{aligned}
f_1 &= 0.25\left(1 + \langle IZ \rangle_\rho + \langle ZI \rangle_\rho + \langle ZZ \rangle_\rho\right) \\
f_2 &= 0.25\left(1 - \langle IZ \rangle_\rho + \langle ZI \rangle_\rho - \langle ZZ \rangle_\rho\right) \\
f_3 &= 0.25\left(1 + \langle IZ \rangle_\rho - \langle ZI \rangle_\rho - \langle ZZ \rangle_\rho\right) \\
f_4 &= 0.25\left(1 - \langle IZ \rangle_\rho - \langle ZI \rangle_\rho + \langle ZZ \rangle_\rho\right).
\end{aligned}
\tag{38}
$$

Thus, it is essential to obtain precise expectation values for the diagonal Pauli operators to improve the accuracy of our estimates. To do this, we use a LocalReadOut scheme from IBM's Qiskit Experiments library (Bravyi et al., 2021). In this scheme we characterize the readout errors of physical qubits on the **FakeBrisbane** backend. These errors are assumed to be local in the sense that they are independent across qubits. Readout error mitigation uses a mitigator object (matrix) computed from an assignment matrix $A$, where each element $A_{i,j}$ represents the probability of observing outcome $i$ when the true outcome is $j$. By applying this mitigator to unmitigated measurement counts, we refine our estimates by obtaining more accurate expectation values for $ZZ$, $ZI$, and $IZ$.

### A.4.2 How and where does it fit in the MAB Routine?

In each round of the MAB policy, based on an Upper Confidence Bound (UCB) score, the sampling rule selects a quantum state to measure. Since only a single-shot measurement is performed per round, the error mitigation procedure described in Section A.4.1 is applied after a state has been measured several times. To illustrate this process, consider a specific round $t = F$, where state $\rho_1$ has previously been measured $T^\star$ times. The unmitigated measurement *counts* for the four possibrle outcomes are denoted as $F_1^{\mathrm{um}}, F_2^{\mathrm{um}}, F_3^{\mathrm{um}}$, and $F_4^{\mathrm{um}}$. The empirical frequencies of these outcomes are given by,

$$
\hat{f}_i^{\mathrm{um}}(F) = \frac{F_i^{\mathrm{um}}}{T^\star}, \quad i \in [4].
\tag{39}
$$

At this point, we invoke the error mitigation routine, supplying it with the unmitigated counts $\{F_i^{\mathrm{um}}\}$ as input. The mitigation routine corrects for readout errors and returns mitigated expectation values of the diagonal Pauli observables, yielding mitigated estimates $\hat{f}_i^{\mathrm{m}}(F)$. With post-processing adjustments to correct for decimal rounding errors, the corresponding mitigated measurement counts,

$$
F_i^{\mathrm{m}} = \hat{f}_i^{\mathrm{m}}(F) \times T^\star, \quad i \in [4].
\tag{40}
$$

We propose a **nested mitigative process** where the MAB algorithm invokes the error mitigation routine once every $F$ measurement shots per state and uses the mitigated values in subsequent shots. For instance, at $t = F$, the routine produces mitigated estimates $\hat{f}_i^{\mathrm{m}}(F)$ from which we obtain mitigated counts. Future measurement outcomes update on these mitigated counts. At $t = 2F$, the routine takes input these new counts and outputs a new set of mitigated estimates $\hat{f}_i^{\mathrm{m}}(2F)$. This creates a nested-mitigation cycle, where each round of mitigation refines the previous one.

We conduct an empirical study to assess the impact of error mitigation on the average copy complexity of the MAB algorithm. Mitigation is invoked once every $F$ rounds, where $F$ ranges from 50 to 10,000 in steps of 50. Here, smaller $F$ values correspond to high-frequency mitigation and larger values indicate lower-frequency mitigation. For the problem instance described in Section 6, with $\delta \in (0,1)$ and range of $F$, we execute Algorithm 3 on FakeBrisbane, averaging the copy complexity at stoppage over 20 runs. The percentage of error mitigation is quantified as the relative reduction in copy complexity compared to the case without mitigation. To ensure the algorithm correctly identifies the entangled states, we employ an error indicator that verifies whether its error remains within the prescribed threshold $\delta$. Using this framework, we generate the heatmap in Fig. 7, which visualizes the percentage reduction in copy complexity due to error mitigation. Notably, the white regions indicate cases where the algorithm converged in finite time but failed to identify

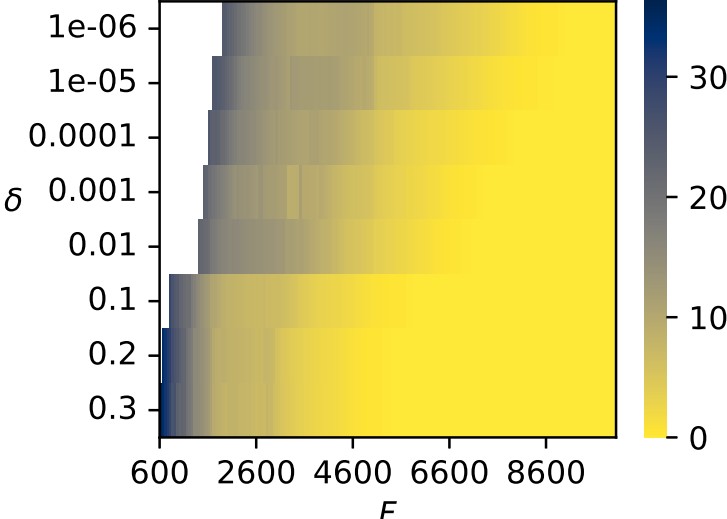

Figure 7: Heatmap of percentage error mitigation on FakeBrisbane backend for $\delta \in (0, 1)$ and various mitigation frequencies

the entangled states correctly. We observe and report the following inferences from Fig. 7. First, the effect of mitigation is $\delta$-dependent. For larger values of $\delta$, the mitigation effect starts only as early as ($F = 600$) and stabilizes faster ($F \sim 4000$). In contrast, for smaller values of $\delta$, the effect of mitigation is prominent only mid-range and stabilizes at $F \sim 7000$. Second, for $F < 600$ and smaller values of $\delta$, the algorithm fails to detect the correct set of states under the prescribed $\delta$. This can be attributed to over-mitigation, which could potentially lead to random fluctuations in the estimates. Third, the observed stabilization zone (yellow) across values of $\delta$ suggests a critical threshold for $F$ beyond which reducing mitigation frequency (increasing the value of $F$) no longer reduces errors. It remains an open question to fully understand and optimize for the use of error-mitigation and integrate them with MAB strategies.

