# OpenReview forum: "Batch Entanglement Detection in Parameterized Qubit States using Classical Bandit Algorithms"
_TMLR — Accepted by TMLR_

### Review · Reviewer_Vty7 · 2025-08-25

**Summary Of Contributions:**

The paper introduces a framework for batch entanglement detection in quantum systems by leveraging techniques from classical multi-armed bandit (MAB) problem. Instead of the full state tomography, the proposed method focuses on detecting entanglement through a targeted set of measurements. Specifically, the authors employ a family of "Witness Basis Measurements" (WBMs) derived from entanglement witnesses, which allow them to compute a separability parameter indicating whether a quantum state is entangled. By mapping each quantum state in a batch to an arm in a bandit model, the detection problem is reformulated as identifying the subset of states for which the separability parameter falls below a certain threshold.

Strength:
- The paper is generally well written, even though too much background material in the beginning of the paper compromises the readability.
- Experimental results using numerical simulations on IBM’s Qiskit platform are provided.

Weakness:
- The main weakness is that this work combines existing literature heavily. In particular, the witness operators measurements and the separability criterion are mostly from Zhu et al. (2010) (in fact, the considered form of witnesses are more confined than that of Zhu et al. (2010)), and the algorithms are from multi-armed bandit community.
- Above is not a critical weakness per se, especially for journals like TMLR. However, there is no evidence regarding if the proposed method is superior to existing methods, nor the scale of the experiments provided confirm the utility of the proposed method.

**Audience:**

No

**Audience Explanation:**

The reviewer generally finds this work to not fit well with TMLR. While some researchers from multi-armed-bandit (MAB) community might find the entanglement detection as an interesting application, this work hardly provides any novelty in terms of MAB. Rather, the claimed novelty lies in applying existing MAB algorithms for batch entanglement detection. In my humble opinion, other venues more tailored for quantum computing / quantum information science would be a much better fit.

**Broader Impact Concerns:**

This work provides a framework for entanglement detection in quantum computing, and there is not much ethical implication to be discussed.

**Claims And Evidence:**

No

**Claims Explanation:**

This work is particularly hard to evaluate, as it utilizes many existing works, both in the multi-armed-bandit (MAB) community as well as entanglement detection community. Here is an exemplary sentence from pg 6: "In HIDoC, the sampling rule is derived from the UCB algorithm for cumulative regret minimization, while the identification rule is based on the LUCB algorithm for BAI, and the APT algorithm for the thresholding bandits problem. The proposed HDoC algorithm (LUCB-G) requires $O( \Delta^{-2} ( K \log 1/\delta + K \log K + K \log 1/\Delta ) )$ samples." The reviewer is unfortunately not an expert in MAB, and thus it is hard to evaluate the novelty of such construction.

Apart from the MAB complication, the provided "Qiskit experiments" consider identifying 3 entangled states out of the 5 candidate states.
Unfortunately, the proposed method is not compared against any existing work, which again hinders evaluating the efficacy of the proposed method.

**Requested Changes:**

Major changes
- Too much background information: the core argument of this paper, MAB for entanglement detection, starts in pg 7. Most of the previous texts are devoted to explaining preliminary concepts. As a result, the readability is quite compromised.
- Comparison against existing methods should be provided. The main experiment seems to concern with identifying 3 entangled states, out of 5; it is hard to gauge the usefulness of the proposed method given the scale of such experiments (full state tomography seems easy in such scale), especially in the absence of comparison to any other algorithm.


Questions
- End of Sec 2.2: "We note that most two-qubit entangled states can be detected under the six witnesses listed in Table 1." What do you mean by "most"? Also is the content of Sec 2.2 is entirely from Zhu et al (2010)?
- What reward is being used for the MAB formulation? What problem is being solved, for a "problem instance denoted by ${\mu}$"?
- Assuming that a unique arm exists -- doesn't this impose some sort of convexity? Is it justified in this framework?
- Below Definition 2: Farrell (1964) already achieves $O( \Delta^{-2} \log \log \Delta^{-2} )$? Then what do the more recent cited works (e.g., Karnin et al. 2013) do?
- Why is the "primary objective" of BAI setting to characterize the expected stopping time $\mathbb{E}_\mu [\tau]$?
- Where is $\mathcal{F}$ defined? (parameterized two-qubit states)
- How does this method compare with simply running the algorithm of Zhu et al. (2010) multiple times?
- How does the theoretical guarantees on the sample complexity compare with existing methods?

Typos
- Top of pg 3: it can represents both pure and mixed states…
- Top of pg 5: The value of S equation 7 depends on …
- Bottom of pg 5: citation styles / notations for lograthims are inconsistent.
- Above Definition 2: $\delta$-PAC? Or $\delta$-PC?
- Pg 15: a sentence stops in the middle after "…, then" before corollary 14

---

> ### Author Response · Authors · 2025-09-04
> **Rebuttal by Authors (1/3)**
>
> We sincerely thank **Reviewer Vty7** for the time and effort invested in reviewing our manuscript. We value the detailed feedback and have carefully addressed each concern below. Before providing point-by-point responses, we wish to emphasise the following:
>
> ### **1. Novelty and Scope**
> To the best of our knowledge, our paper is the first to connect entanglement detection with multi-armed bandit algorithms, yielding order-optimal copy complexity guarantees. Well-known entanglement detection methods, such as full-scale tomography (FST) and fixed witness testing, lack adaptivity and do not provide explicit confidence intervals and bounds on copy complexity for entanglement detection. Thus, our contribution fills this crucial gap.
>
> ### **2. Practical Impact**
> Each “pull’’ corresponds to consuming a fresh quantum state copy, which, as an experimental resource, is scarce. By minimising the ‘stopping time’ of the algorithm, the framework directly aims to reduce the copy complexity given a batch of parameterised bipartite qubit states. We validate this by conducting experiments that involve constructing quantum circuits for state generation and witness-based measurements, and subsequently simulating them on IBM Quantum cloud, demonstrating practical feasibility under realistic noise.
>
> ### **3. Interdisciplinary Significance**
> The work bridges classical machine learning and quantum entanglement theory, thereby introducing a principled framework for adaptive entanglement detection.
>
> ### **4. Why TMLR?**
> The transition from classical to quantum involves classical-quantum collaborations, wherein classical concepts are used to solve quantum information problems and vice versa. Our work proposes the use of classical machine learning algorithms in conjunction with specific quantum measurements to address the significant problem of entanglement detection. More broadly, the spirit of this work lies in `quantum-classical co-design'. For the foreseeable future, we expect quantum systems to operate in a non-stand-alone mode alongside classical systems, where quantum systems operate with feedback from classical systems.
>
> ### **Remark 1: does it work beyond $(m,K)=(3,5)$?**
>
> **Response:**  Our method is not restricted to this case. The experimental illustration with $K = 5$ and $m = 3$ was chosen for the purpose of explanation. Our method identifies any subset of $m$ entangled states out of $K$, with neither prior knowledge of which states are entangled nor the number of entangled states $m$. This is reflected in the $(m, K)$-quantum multi-armed bandit framework formalised in Section 3 of our paper.
>
> ### **Remark 2: Too much background**
>
> **Response:**  Since the work bridges classical machine learning and quantum entanglement detection, we included preliminaries to make the paper self-contained for both communities. We will streamline these sections to focus on the essential content and incorporate it into the revised manuscript (to be uploaded).
>
> ### **Remark 3: End of Sec 2.2: "We note that most two-qubit entangled... content of Sec 2.2 is entirely from Zhu et al (2010)?**
>
> **Response:**  We would like to clarify that the phrase "most two-qubit entangled states" refers to the fact that the six entanglement witnesses introduced by Zhu et al. (2010) are able to detect a large majority, though not the entirety, of entangled two-qubit states. All entangled states detectable by the six witnesses in Zhu et al. (2010) are detectable by our proposed mechanism. Section 2.2 is a preliminary section explaining the six witnesses proposed by Zhu et al. (2010).
>
> ### **Remark 4: Reward and problem instance**
>
> **Response:**  In our setting, each of the six entanglement witnesses is associated with a witness–basis measurement (WBM), which decomposes into four projective elements. We fix a WBM $\mathcal{E}$ and then measure the witness on a candidate state $\rho$ (i.e., single pull of an arm) and that yields one of these four outcomes, distributed according to Born’s rule. The measurement data (i.e., reward in MAB) is therefore the frequency with which each outcome is observed across repeated pulls, and from these empirical frequencies we construct the separability criterion $S_\mathcal{E}(\rho)$. The collection of $\mathbf{S}_{\mathcal{E}}(\cdot)$ across all the $K$ states in the ensemble constitutes a problem instance $\boldsymbol\mu$ in the bandit formulation. We refer the reviewer to Table 2 of our draft for further clarification.
>
> ### **Remark 5: Where is $\mathcal{F}$ defined? (parameterized two-qubit states)**
>
> **Response:** The notation $\mathcal{F}$ is used to denote the set of parameterised two-qubit states, namely, Werner, Bell-Diagonal, and Noisy Bell states, which are detectable under the first two witnesses in Zhu et al (2010). We refer the reviewer to page 8, line 5, where the notation $\mathcal{F}$ was introduced and will clarify this explicitly in the revised manuscript (to be uploaded).

---

> ### Author Response · Authors · 2025-11-03
> **Rebuttal by Authors (2/3)**
>
> ### **Remark 6: Assuming that a unique arm exists -- doesn't this impose some sort of convexity? Is it justified in this framework?**
> **Response:** The authors are unclear what the reviewer means by convexity and request clarification. The assumption of a unique best arm is standard in MAB formulations to simplify guarantees (for e.g., [see this paper](https://arxiv.org/abs/1602.04589)). However, our results hold in the general $(m, K)$ good-arm identification setting, where uniqueness is not required. We refer the reviewer to Section 2.3.2 for further details. In the case when $m=0$, the algorithm returns the correct result, i.e., that no states are entangled with probability at least $1-\delta$, where $\delta$ is the probability of error.
>
> ### **Remark 7: Farrell (1964) already achieves $O( \Delta^{-2} \log \log \Delta^{-2} )$ Then what do the more recent cited works (e.g., Karnin et al. 2013) do?**
> **Response:**  We note that Farrell’s bound provides the asymptotic optimal rate $O(\Delta^{-2}\log\log \Delta^{-2})$. An asymptotic optimal rate means that the algorithm learns at the quickest achievable speed (theoretically) in the small-error limit and matches the prescribed information-theoretic lower bounds up to a multiplicative factor. Recent works (Karnin et al., 2013; Jamieson et al., 2014) refine this in finite-sample regimes and propose practically implementable algorithms in real-time that achieve near-optimal sample complexity with explicit constants and stopping rules. Our work builds directly on these finite-sample, implementable algorithms.
>
> ### **Remark 8: It is hard to gauge the usefulness of the proposed result given the scale of such experiments (full state tomography seems easy in such scale).?**
> **Response:** The main advantage of our approach lies in its ability to significantly minimise the copy complexity of batch entanglement detection compared to full state tomography (FST) and provides rigorous guarantees on whether the set of $m$ entangled states (out of $K$) is identified correctly. We clarify these two aspects as follows:
> ### **Fixed-confidence guarantees**
> The proposed MAB policies for batch entanglement detection are $(0, \delta)$-PAC. In other words, they provide *exact correctness* $(\epsilon = 0)$ and *guarantee with high probability* $1-\delta$ that the outputted $m$ states are entangled. On the contrary, FST yields a state reconstruction but **does not** provide any explicit confidence guarantee on the separability aspect. Let us consider an example of a Depolarized Bell state, $$
> \rho(w) = (1-w) |{\Phi^+}\rangle\langle{\Phi^+}| + w\,\frac{\mathbb{I}}{4},
> $$ which is separable when $-1/3 \le w \le 1/3$ and entangled when $1/3 < w \le 1$. Let us consider a separable state with parameter $w = 0.32$. To estimate $\hat{w}$ up to a statistical uncertainty $\pm 0.02$ using FST will require $\mathcal{O} (10^4)$ samples to reconstruct the density matrix. This leads to small fluctuations of the estimate $\hat{w} = 0.30$--$0.34$ and may shift the reconstructed state into the $w > 1/3$ region in case of positive error, leading to a false-positive entanglement detection. The proposed MAB approach, however, uses confidence bounds on the separability criterion values and certifies entanglement only when the entire confidence interval lies below the threshold 0. This means that even if a misclassification occurs, its probability is explicitly bounded by $\delta$, thereby ensuring that the policy's outcome is statistically accounted for.
> ### **Empirical Evidence – Kindly refer to Fig.4 of our paper**
> For the batch detection among parameterised bipartite states, FST must be done for *each* state since one does not know which of the $K$ states are entangled. To do FST up to a specified trace-distance accuracy of $\epsilon^\prime$ with collective measurements requires $O(K \times 16/\epsilon^{\prime^2})$ copies ([see this paper](https://arxiv.org/abs/1508.01797) for more details). At the scale of our experiments, FST requires in $\mathcal{O}(10^6)$ total copies to achieve an accuracy as small as $\epsilon^\prime \approx 10^{-2}$, while the MAB approach identifies all entangled states with exact correctness and high probability using only $\mathcal{O}(10^4)$ copies per state across various IBMQ backends like Aer/FakeBrisbane/ibm-brisbane. Quantitatively, the proposed MAB framework demonstrates a clear advantage, achieving a reduction of up to two orders of magnitude in copy complexity compared to FST.
> In conclusion, our approach offers a quantitatively demonstrated reduction in copy complexity, explicit confidence guarantees, and scalability for batch detection tasks in parameterised bipartite qubit states.

---

> ### Author Response · Authors · 2025-11-03
> **Rebuttal by Authors (3/3)**
>
> ### **Remark 9: Why is the "primary objective" of BAI setting to characterize the expected stopping time $\mathbb{E}_\mu [\tau]$?**
> **Response:** One of the objectives in the BAI setting is to characterise the expected stopping time $\mathbb{E}_{\boldsymbol\mu}[\tau]$. This is a performance metric for evaluating sample efficiency under fixed-confidence guarantees. In other words, a margin of error $\delta$ is fixed and the goal of the policy is to identify the best arm or good arms by using as few samples with high probability $1-\delta$. In our paper, an arm pull corresponds to measuring a witness on a fresh copy of the candidate state. Thus, the stopping time in the bandit setting is equal to the number of physical state copies consumed in the quantum setting, which is the central resource bottleneck. Hence, copy complexity and stopping time are equivalent, quantifying the performance of the policy.
>
> ### **Remark 10: How does this method compare with simply running the algorithm of Zhu et al. (2010) multiple times?**
> **Response:** Running multiple iterations of Zhu et al. (2010)'s witness testing approach corresponds to sequential single-state tests using repeated witness measurements and **without adaptive resource allocation** across the $K$ states and no definite stopping rule. While repetition can reduce statistical noise in the $S_{\mathcal{E}}$ estimates, it suffers from a limitation similar to full-state tomography (FST): both approaches treat each state independently and expend unnecessary copies—FST toward exhaustive state reconstruction, and repetitive applications of Zhu et al. (2010) toward non-adaptive measurement repetition. In contrast, the proposed MAB approach allocates measurements adaptively across the $K$ states. It focuses measurement effort on states with higher uncertainty in their $S_{\mathcal{E}}$ estimates and stops early for those states that can be confidently classified. This adaptivity reduces total copy consumption and also provides fixed-confidence guarantees, which are absent in the other approaches like FST or repeated fixed witness testing.
>
> ### **Remark 11: How does the theoretical guarantees on the sample complexity compare with existing methods?**
> **Response:** The existing approaches to entanglement detection are as follows:
> - **FST:** Kindly refer to the response to Remark 2. This can be followed by verifying the PPT criterion. In other words, one needs to verify whether the partial transpose of the reconstructed state has a negative eigenvalue to certify entanglement.
> - **Fixed witnesses (Zhu et al. 2010):** Measure a family of witnesses repeatedly until the statistical error is minimised. Here, since witness measurement outcomes are bounded, one can employ Hoeffding-style bounds on the estimates. However, there is no adaptive stopping rule to help optimise when to stop measuring, and a finite fraction of entangled states go undetected unless FST is invoked.
>
> In our proposed solution, we retain the same six-witness family as the detection primitive but recast the allocation as a $(m,K)$ Thresholding Bandit problem. Firstly, Theorem 10 shows that the truly entangled states in an ensemble of $K$ states are identified using $\sum_{i=1}^K O \Big(\Delta_i^{-2}\log\tfrac{K\log \Delta_i^{-2}}{\delta}\Big)$ copies, within logarithmic factors of the information-theoretic lower bound. Corollary 14 further refines this to $O(\Delta^{-2}(K\log(1/\delta)+K\log K+K\log\log 1/\Delta))$, exhibiting the  linear dependence on $\log(1/\delta)$. Secondly, elimination-based policies stop early on fairly easily detectable states and concentrate resources on those states that are harder to distinguish, i.e., those close to the threshold. Thirdly, Fig. 4 empirically certifies that detecting entanglement across $K=5$ states requires only a few $10^4$ copies, two orders of magnitude fewer than FST and fixed-witness repetition.
>
> ### **Remark 12: Typos**
> **Response:** We acknowledge all the typographical inconsistencies in our paper and will incorporate the necessary changes in the revised manuscript (to be uploaded).

---

### Review · Reviewer_AdQr · 2025-10-11

**Summary Of Contributions:**

The paper addresses the batch entanglement detection problem, which aims to identify all entangled states from a batch of K unknown quantum states without performing full state tomography efficiently. The main contribution is establishing a connection between this quantum problem and the classical Thresholding Bandit Problem (TBP) in Multi-Armed Bandits (MAB). The authors propose a so-called (m, K)-quantum MAB framework where each quantum state is an "arm" and a separability criterion derived from a family of six WBMs serves as the parameter to be estimated. They apply classical MAB policies like Successive Elimination and lil'HDoC to this framework, demonstrating that they can efficiently identify entangled states with high probability and with theoretical guarantees on sample complexity. The approach is verified through numerical simulations on parameterized two-qubit states and real quantum hardware experiments, as well as on arbitrary quantum states.

**Audience:**

Yes

**Audience Explanation:**

The paper's novelty lies in drawing a connection between batch entanglement detection and a Thresholding Bandit problem in classical MAB. This demonstrates the potential for employing classical machine learning techniques to solve complex problems in quantum mechanics. This direct application of classical ML theory to quantum physics is a rapidly growing field of interest.

**Broader Impact Concerns:**

The paper focuses on an efficient technical method for detecting quantum entanglement. No Broader Impact Statement section is required.

**Claims And Evidence:**

Yes

**Claims Explanation:**

The claims made in the submission are supported by evidence, which is presented clearly through a combination of theoretical analysis, numerical simulations, and real-world quantum hardware experiments.

**Requested Changes:**

Here are some suggestions to improve the conceptual clarity and rigor.

1. Can the authors summarize in the introduction how the MAB framework overcomes the exponential complexity of FST?

2. Summarize the exponential measurement complexity of FST into a lemma to compare with Theorems 10 and 13.

3. Add the ablation to discuss the algorithm's sensitivity to hyperparameters, such as $\epsilon$ and T.

---

> ### Author Response · Authors · 2025-11-03
> **Rebuttal by Authors**
>
> We sincerely thank the **Reviewer AdQr** for the time and effort invested in reviewing our manuscript. We value the detailed feedback and have carefully addressed each concern below.
>
> **Remark 1: Can the authors summarise in the introduction how the MAB framework overcomes the exponential complexity of FST?**
>
> **Response:** Full state tomography (FST) requires linearly independent measurements that scale exponentially with the number of qubits. For the bipartite qubit case, FST reconstructs the entire $4\times4$ density matrix using $\mathcal{O}(16/\epsilon^2)$ copies per state to achieve an $\epsilon$-accurate estimate in trace distance, resulting in $\mathcal{O}(K\times16/\epsilon^2)$ total copies for $K$ states. By contrast, the MAB policies in our proposed approach are $(0,\delta)$-PAC, meaning they correctly identify the set of entangled states with probability at least $1-\delta$. These policies adaptively allocate the measurement effort to states with high uncertainty in the $S_\mathcal{E}$ estimate and do not allocate for states that can be classified confidently. This type of allocation of copies across states avoids redundant reconstruction of the state. Secondly, there is empirical advantage as suggested by our experiments (see \textbf{Fig.~4} of our paper), utilizing $\mathcal{O}(10^4)$ copies amounting up to two orders of magnitude fewer copies than performing FST with accuracy $\mathcal{O}(10^{-2})$, while also ensuring that any residual uncertainty is explicitly bounded by $\delta$. We will incorporate these details into the revised manuscript (to be uploaded) to clarify how the MAB framework can be more sample-efficient compared to FST.
>
> **Remark 2: Summarize the exponential measurement complexity of FST into a lemma to compare with Theorems 10 and 13.**
>
> **Response:** For bipartite qubits ($d = 4$), FST reconstructs the state by estimating $15$ parameters of the density matrix to within a trace-distance accuracy of ~$\epsilon$ using $\mathcal{O}(16/\epsilon^2)$ copies with optimal collective measurements
> ([see this paper](https://arxiv.org/abs/1508.01797)). For $K$ candidate states, this will incur $\mathcal{O}(16K/\epsilon^2)$ copies. The copy complexity under the proposed MAB approach (Theorems 10 and 13) scales polynomially in $K$ and logarithmically in $1/\delta$ while the FST copy complexity for the batch entanglement detection problem scales linearly with $K$. We will state this precisely in the revised manuscript (to be uploaded).
>
> **Remark 3: Add the ablation to discuss the algorithm's sensitivity to hyperparameters, such as $\epsilon$ and $T$.**
>
> **Response:** The parameter $\epsilon$ appears in the LIL-based confidence radius expression stated in Lemma 7, page 11 of our paper. We note that small values of $\epsilon$ tighten the confidence radius and therefore incur more samples before elimination, while large $\epsilon$ values reduce the number of samples with a higher chance of premature arm elimination. The asymptotic growth of $U(t,\delta)$ with $\epsilon$ is sublinear and the policy's correctness remains unaffected for $\epsilon > 0$. The warm-start phase parameter $T$ controls the number of measurements collected before adaptive allocation begins. If $T=1$, the algorithm enters the adaptive phase with a large confidence radius $U(1,\delta)$; while algorithmic correctness is preserved, early rounds may take longer to stabilise. A moderate value of $T$ as given in Eq. (15) of our paper provides a copy complexity that follows from Theorems 10 and 13 in our paper. If $T$ is larger than the typical stopping time $\tau^\star$, extra measurements get wasted, linearly degrading as $(T-\tau^\star)_+$ while preserving algorithmic correctness. Hence, $T$ governs the onset of the LIL regime; once this threshold is crossed, the $(0,\delta)$–PAC guarantees and copy complexity depend primarily on the sub-optimal gaps $\{\Delta_i\}$, number of parameterised batch states $K$ and the error margin $\delta$. We will incorporate these changes in the revised manuscript (to be uploaded).

---

> > ### Comment · Reviewer_AdQr · 2025-12-25
> >
> > Thanks for your response. My concerns have been addressed. It is great to incorporate the response into the revision.
> >
> > Best,
> > Reviewer AdQr

---

### Review · Reviewer_muTT · 2025-10-25

**Summary Of Contributions:**

The paper tackles the problem of efficiently detecting which quantum states in a given batch are entangled, without performing full quantum state tomography (FST) or adaptive measurements.
In quantum information processing, such as quantum communication, computation, or teleportation, many qubit pairs are produced, and it is crucial to verify which of them are genuinely entangled.
However, existing methods for entanglement detection have significant limitations:
1) Full-state tomography requires exponentially many measurements as the qubit number increases.
2) Individual entanglement witnesses can only detect specific classes of entangled states, not all possible ones.
To address these challenges, the authors formulate the batch entangled state detection problem as a Good Arm Identification (GAI) problem in the multi-armed bandit framework.
Each quantum state corresponds to an arm, and the goal is to identify which arms (states) are "good" (entangled) based on noisy measurement outcomes.
The authors propose two algorithms, Successive Elimination for exact one entangled state detection and lil'sHDoC for multiple entangled states without prior knowledge of their number.
The author provides theoretical guarantees on the sample complexity of these algorithms, showing they are $\delta$-PAC (probably approximately correct) algorithms.
Authors also conduct numerical simulations to validate the performance of their algorithms, demonstrating significant reductions in the number of measurements required compared to naive approaches.

**Additional Comments:**

I am more familiar with the multi-armed bandit literature than quantum information theory. From the MAB perspective, I am glad to see the authors provide theoretical guarantees and empirical validation of their algorithms.
For the quantum information side, while I am not an expert, the problem of batch entanglement detection seems relevant and important.
I hope the authors can address the clarity and presentation issues mentioned above to make the paper more accessible to a broader audience.
The interdisciplinary nature of the work is a strength, and I believe it will inspire further research at the intersection of quantum information and classical learning theory.

**Audience:**

Yes

**Audience Explanation:**

Formalizing Qubit States as a multi-armed bandit is an interesting topics. Researchers have developed many MAB algorithms on abstract environments for a long time. Finding a good application is good for the community of MAB.

**Broader Impact Concerns:**

This work formalizes a new application of Multi-armed Bandit (MAB) problem in the Qubit States detection. I believe this will broader the application of MAB algorithms in the long turn.

**Claims And Evidence:**

Yes

**Claims Explanation:**

Strengths:

1) The paper presents a novel intersection between quantum information theory and classical multi-armed bandit (MAB) frameworks. This interdisciplinary approach is innovative and has the potential to inspire further research at the intersection of these fields. Compared to the previous works, such as Lumbreras et al. (2022) which works on a global entanglement detection, this work focuses on batch entanglement detection, which is more practical in many quantum information processing scenarios.

2) The author provides a rigorous theoretical analysis of the proposed algorithms, including sample complexity bounds and PAC guarantees. This adds significant value to the work, as it proposes new algorithms with strong theoretical guarantees of their performance.

3) The authors demonstrate feasibility through IBM Qiskit simulations and hardware tests (ibm-brisbane backend). The inclusion of real-device results, noise considerations, and performance trends with respect to error probability $\delta$ substantially strengthens the empirical credibility of the framework.

Weaknesses:

1) The writing could be condensed in places; Sections 2 and 3 span too long with textbook-level exposition before contributions emerge. It would be beneficial to streamline these sections to focus more on the novel contributions of the paper and how you formalize the entangled state detection problem as a GAI problem. This would help readers quickly grasp the significance of your work.

2) The author mentions that the proposed algorithms outperform naive approaches in numerical simulations. It would be good to provide a fair comparison with existing state-of-the-art methods for entanglement detection, if applicable. This would help to contextualize the performance improvements and demonstrate the practical advantages of your algorithms.

**Requested Changes:**

1: Condense the writing:
The background on entanglement witnesses, separability criteria, and stochastic bandits is lengthy and partly textbook-like. Consider moving derivations and detailed examples to an appendix and summarizing the essentials in 1-2 paragraphs within the main text.
Merge or clearly separate theoretical versus experimental parts. Sections 3 and 4 could start with short “Problem-Goal-Contribution” paragraphs to orient the reader. Include a short subsection or figure summarizing how the “quantum reward model” corresponds to classical MAB components.

2: Clarify assumptions on witness basis measurements:
Explicitly state that the witness basis measurements (WBMs) are known and fixed. Discuss how the method might extend if the WBM were partially unknown or noisy.

---

> ### Author Response · Authors · 2025-11-03
> **Rebuttal by Authors (1/2)**
>
> We sincerely thank the **Reviewer muTT** for the time and effort invested in reviewing our manuscript. We value the detailed feedback and have carefully addressed each concern below.
>
> **Remark 1: Editorial changes to condense writing for presentation.**
>
> **Response:** Since the work bridges classical machine learning and quantum entanglement detection, we included preliminaries to make the paper self-contained for both communities. We will streamline Section 2 to focus on the essential preliminary content required for this paper and will clarify the “Problem–Goal–Contribution” pipeline to better orient the reader. We will incorporate it into the revised manuscript (to be uploaded).
>
> **Remark 2: The author mentions that the proposed algorithms outperform naive approaches in numerical simulations. It would be good to provide a fair comparison with existing state-of-the-art methods for entanglement detection, if applicable. This would help to contextualize the performance improvements and demonstrate the practical advantages of your algorithms.**
>
> **Response:** Two existing approaches for entanglement detection:
> - **Full State Tomography (FST) for bipartite qubit states followed by PPT test**: For the batch detection among parameterised bipartite states, FST must be done for \textit{each} state since one does not know which of the $K$ states are entangled. To do FST up to a specified trace-distance accuracy of $\epsilon^\prime$ with optimal collective measurements requires $O( 16K/\epsilon^{\prime^2})$ copies ([see this paper](https://arxiv.org/abs/1508.01797)). One can verify the PPT criterion for entanglement, that is, check if the partial transpose of the reconstructed state yields a negative eigenvalue. While FST yields an approximate state reconstruction, it **does not** provide any explicit confidence guarantee on the separability aspect.
> - **Fixed witness testing as proposed by Zhu et al. (2010)**: Running multiple iterations of Zhu et al. (2010)'s witness testing approach corresponds to sequential single-state tests using repeated witness measurements and **without adaptive resource allocation** across the $K$ states and no definite stopping rule. While repetition can reduce statistical noise in the $S_{\mathcal{E}}$ estimates, it suffers from a limitation similar to full-state tomography (FST): both approaches treat each state independently and expend unnecessary copies—FST toward exhaustive state reconstruction, and repetitive applications of Zhu et~al. (2010) toward non-adaptive measurement repetition.
>
> Firstly, the proposed MAB policies for batch entanglement detection provide exact correctness $(\epsilon = 0)$ and guarantee with high probability $1-\delta$ that the outputted $m$ states are entangled. These policies use confidence bounds on the separability criterion values and certify entanglement only when the entire confidence interval lies below the threshold 0. This means that even if a stately is falsely classified to be entangled, its probability is explicitly bounded by $\delta$,  ensuring that the policy's outcome is statistically accounted for. On the contrary, the above two approaches do not provide any guarantees on the correctness of the solution.  Secondly, the MAB policies allocate measurements adaptively across the $K$ states by focusing measurement effort on states with higher uncertainty in their $S_{\mathcal{E}}$ estimates and stops early for those states that can be confidently classified. This adaptivity is absent in the above two approaches since the measurements proceed sequentially for every state with no explicit stopping rule. Thirdly, at the scale of our experiments, FST requires in $\mathcal{O}(10^6)$ total copies to achieve an accuracy as small as $\epsilon \approx 10^{-2}$, while the MAB approach identifies all entangled states with exact correctness and high probability using only $\mathcal{O}(10^4)$ copies per state across various IBMQ backends like Aer/FakeBrisbane/ibm-brisbane (Kindly refer to Fig. 4 of our paper). Quantitatively, the proposed MAB framework demonstrates a clear advantage, achieving a reduction of up to two orders of magnitude in copy complexity compared to FST. In conclusion, our approach offers a quantitatively demonstrated reduction in copy complexity, explicit confidence guarantees, adaptivity in distributing measurement effort and scalability for batch detection tasks. We will streamline these details carefully in our revised manuscript (to be uploaded).

---

> ### Author Response · Authors · 2025-11-03
> **Rebuttal by Authors (2/2)**
>
> **Remark 3: Clarify assumptions on witness basis measurements: Explicitly state that the witness basis measurements (WBMs) are known and fixed. Discuss how the method might extend if the WBM were partially unknown or noisy.**
>
> **Response:** We will explicitly mention in the revised manuscript (to be uploaded) that the WBMs used in this paper are **known and fixed** and the states are unknown. Additionally, we will state that our proposed MAB approach assumes access to the exact projective forms of the WBM. If the WBM is partially unknown or noisy, the measurement uncertainty should be accounted for. This would require a separate WBM calibration stage prior to adaptively allocating measurements across $K$ states. While this still preserves the $(0,\delta)$-PAC guarantees, the measurement uncertainty could inflate the MAB confidence width $U(t,\delta)$, thereby incurring more samples.

---

### Public Comment · ~Ajay_Narayanan_Sridhar1 · 2025-09-02

I’m not an expert in quantum physics, but I’m familiar with multi-armed bandits and thresholding problems. I found it really interesting to see entanglement detection framed through that lens. The idea of treating each quantum state as an arm and applying bandit-style sampling feels very natural from a learning perspective. It’s impressive that standard algorithms like Successive Elimination and lil’HDoC can be adapted here. Even without deep knowledge of the quantum side, the approach makes the problem feel intuitive and well-structured. I also appreciate that the paper includes experiments on real quantum hardware. Curious to see how this method might scale beyond two-qubit systems. Great work overall.

---

### Decision · Action_Editor_ecpj · 2025-12-25

**Recommendation:** Accept as is

**Audience:**

Yes

**Audience Explanation:**

Though TMLR audience perhaps does not have large number of quantum computing/information theorists, this paper can be seen as an application of ML to a new domain. There is a substantial number of members in TMLR community who will be interested in application of multi-arm bandits to other domains.

**Claims And Evidence:**

Yes

**Claims Explanation:**

This paper uses techniques from multi-arm bandits literature to detect quantum entanglement among states. This is an interesting approach to batch entanglement detection that uses a classical online learning approach. The theorems and claims appears correct; and the experimental validations sufficient for publication.